# Aligned Multi Objective Optimization

**Yonathan Efroni** [1]  **Ben Kretzu** [2]  **Daniel R. Jiang** [1]  **Jalaj Bhandari** [1]  **Zheqing Zhu** [1]  **Karen Ullrich** [1]

## Abstract

To date, the multi-objective optimization literature has mainly focused on conflicting objectives, studying the Pareto front, or requiring users to balance tradeoffs. Yet, in machine learning practice, there are many scenarios where such conflict does not take place. Recent findings from multi-task learning, reinforcement learning, and LLMs training show that diverse related tasks can enhance performance across objectives simultaneously. Despite this evidence, such phenomenon has not been examined from an optimization perspective. This leads to a lack of generic gradient-based methods that can scale to scenarios with a large number of related objectives. To address this gap, we introduce the Aligned Multi-Objective Optimization framework, propose new algorithms for this setting, and provide theoretical guarantees of their superior performance compared to naive approaches.

## 1. Introduction

In many real-world optimization problems, we have access to multi-dimensional feedback rather than a single scalar objective. The multi-objective optimization (MOO) literature has largely focused on the setting where these objectives *conflict* with each other, which necessitates the *Pareto dominance* notion of optimality. A closely related area of study is *multi-task learning* (Caruana, 1997; Teh et al., 2017; Sener and Koltun, 2018; Yu et al., 2020; Lin et al., 2021; Liu et al., 2021; Navon et al., 2022; Zhou et al., 2022; Lin and Zhang, 2023; Liu et al., 2023a; Chen et al., 2023; Achituve et al., 2024; He et al., 2024; Liu and Vicente, 2024), where multiple tasks are learned jointly, typically with both shared and task-specific parameters. The hope is that the model can perform better on individual tasks by sharing common information across tasks. Indeed, the phenomenon of im-

proved performance across all tasks has been observed in several settings (Lin and Zhang, 2023; Lee et al., 2024), suggesting that perhaps there may not always be significant trade-offs between objectives. Similar observations appear in meta-learning (Ravi and Larochelle, 2017; Finn et al., 2017; Hospedales et al., 2021), where the goal is to learn representations that enable quick adaptation to new tasks with minimal additional training, as well as in reinforcement learning (Jaderberg et al., 2016; Teh et al., 2017; Veeriah et al., 2019; Dann et al., 2023), where practitioners use multiple reward functions to better specify the policy or its representation.

In this work, we explicitly study a setting where objectives are *aligned*, namely, that the different objectives share a common solution. This situation arises frequently in practice. For example, when using reinforcement learning (RL) to augment large language models (LLMs) with reasoning capabilities, there are often multiple options for the choice of reward model to use. Lightman et al. (2023) and Uesato et al. (2022) consider both outcome and process-based rewards, and, recently Guo et al. (2025); Team et al. (2025) discuss the use of accuracy, format rewards, length of response, and reward on math problems as additional reward functions. In training text-to-image models using RL, Lee et al. (2024) use four reward models (aesthetic quality, human preference, text-image alignment, and image sentiment) and show results where *all* rewards are increased. Although the method of Lee et al. (2024) is designed for finding Pareto-optimal solutions (implying the existence of trade-offs), the numerical results suggest that the objectives may actually be *aligned* to a good degree.

These observations are also related to a more general phenomenon in RL discussed by Dann et al. (2023), where learning can be accelerated by exploiting several alternative reward specifications that all lead to the same optimal policy. This concept builds on prior work showing that the choice of reward function (e.g., dense versus sparse reward) can have a dramatic effect on training time (Ng et al., 1999; Luo et al., 2020; Wang et al., 2019; Hu et al., 2020). A related idea in statistics is that when labeled data is sparse, practitioners can rely on closely-related proxy tasks to improve prediction accuracy (Bastani, 2021).

To our knowledge, there is no work that studies such a

---

*Equal contribution  [1]Meta AI  [2]Technion. Correspondence to: Yonathan Efroni <yonathane@meta.com>.

*Proceedings of the 42nd International Conference on Machine Learning*, Vancouver, Canada. PMLR 267, 2025. Copyright 2025 by the author(s).

| Algorithm | Asymptotic Convergence |
|:---:|:---:|
| CAMOO | $O\left((1 - \mu_{\mathrm{G}}/\beta)^k\right)$ |
| PAMOO | $O\left((1 - \mu_{\mathrm{L}}/\beta)^k\right)$ |

Table 1: The main results introduced in this work. $\beta$ is the smoothness parameter; both $\mu_{\mathrm{G}}$ and $\mu_{\mathrm{L}}$ are structural quantities introduced in Section 4 and Section 5. These characterize notions of optimal curvature of weighted function and satisfy $\mu_{\mathrm{L}} \geq \mu_{\mathrm{G}}$.

framework from an optimization perspective. We ask the following question:

*Can gradient descent type of algorithms benefit from multi-objective feedback when the objectives are aligned?*

Previous work in multitask learning had provided convergence guarantees for gradient descent-type algorithms for MOO (Sener and Koltun, 2018; Yu et al., 2020; Liu et al., 2021; Navon et al., 2022; He et al., 2024). However, since these consider general multi-objective framework, their algorithms converge with worst-case guarantees with no meaningful convergence improvement of MOO.

We provide a positive answer to the aforementioned question. We formally introduce the *aligned multi-objective optimization* (AMOO) framework. Subsequently, we design new gradient descent-type algorithms and establish their provable improved convergence in the AMOO setting. These can be interpreted as parameter-free algorithms to handle multi-objective feedback when objectives are aligned. Lastly, we conclude by providing empirical evidence of the improved convergence properties of the new algorithms.

## 2. Related Work

### 2.1. Gradient Weights in Multi-task & Meta Learning

Our work is closely related to optimization methods from the multi-task learning (MTL) and meta learning literature, particularly those that integrate weights into the task gradients or losses. The *multiple gradient descent algorithm* (MGDA) approach of Fliege and Svaiter (2000); Désidéri (2012); Sener and Koltun (2018); Zhang et al. (2024) is one of the first works along this direction. It proposes an optimization objective that gives rise to a weight vector that implies a descent direction for all tasks and converges to a point on the Pareto set. The PCGrad paper (Yu et al., 2020) identified that conflicting gradients can be detrimental to MTL. The authors then propose to modify the gradients to remove this conflict (by projecting each task's gradient to the normal plane of another task), forming the basis for the PCGrad algorithm. Another work that tackles conflicting gradients is the *conflict-averse gradient descent* (CAGrad)

method of (Liu et al., 2021). CAGrad generalizes MGDA: its main idea is to minimize a notion of "conflict" between gradients from different tasks, while staying nearby the gradient of the average loss. Notably, CAGrad maintains convergence toward a minimum of the average loss. Another way to handle gradient conflicts is the Nash-MTL method of Navon et al. (2022), where the gradients are combined using a bargaining game. Very recently, Achituve et al. (2024) introduced a Bayesian approach for gradient aggregation by incorporating uncertainty in gradient dimensions. Other optimization techniques for MTL include tuning gradient magnitudes so that all tasks train at a similar rate (Chen et al., 2018), taking the geometric mean of task losses (Chennupati et al., 2019), and random weighting (Lin et al., 2021). On the meta learning front, the MAML algorithm (Finn et al., 2019) aims to learn a useful representation such that the model can adapt to new tasks with only a small number of training samples. Since fast adaptation is the primary goal in meta learning, MAML's loss calculation differs from those found in MTL.

Few prior works provided provable convergence guarantees of the different existing multi-objective optimization methods. Without additional assumption on the alignment of different objectives, these guarantees quantified convergence to a point on the Pareto front. Unlike our work, there the convergence guarantees depended on worst-case structural quantities such as the maximal Lipschitz constant among all objectives (Liu et al., 2021; Navon et al., 2022; Zhou et al., 2022; Chen et al., 2023; Liu and Vicente, 2024) or the maximal generalized smoothness (Zhang et al., 2024).

The algorithms introduced in the following are similar to existing ones in that they construct a weighted loss to combine information from different sources of feedback. Unlike previous work, we focus on exploiting the prior knowledge that the objectives are *aligned*. We introduce new instance-dependent structural quantities that reflect how aligned multi-objective feedback can improve GD performance, improving convergence that depends on worst-case structural quantities, as in prior works.

### 2.2. Proxy & Multi-fidelity Feedback

Other streams of related work are (1) machine learning using proxies and (2) multi-fidelity optimization. These works stand out from MTL in that they both focus on using *closely related* objectives, while traditional MTL typically considers a set of tasks that are more varied in nature. Proxy-based machine learning attempts to approximate the solution of a primary "gold" task (for which data is expensive or sparsely available) by making use of a proxy task where data is more abundant (Bastani, 2021; Dzyabura et al., 2019). Similarly, multi-fidelity optimization makes use of data sources of varying levels of accuracy (and potentially lower com-

The Specification Example

The Selection Example

The Local Curvature Example

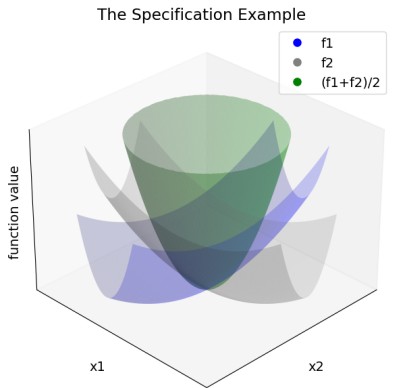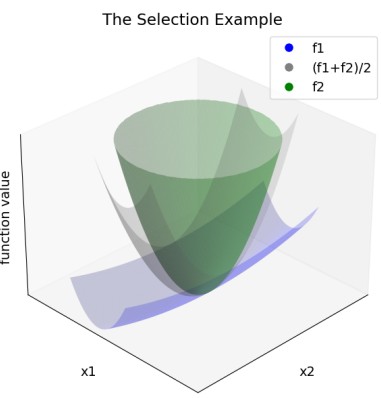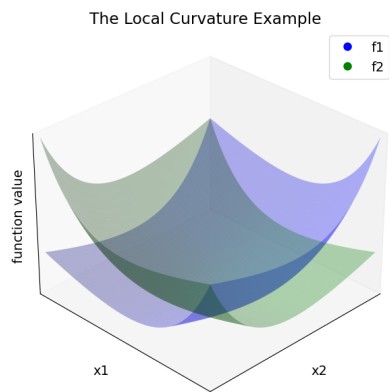

Figure 1: Visualization of AMOO instances in which it is possible to obtain improved convergence compared to optimizing individual functions or the average function: **(left)** the specification example, **(center)** simpler instance of the selection example, and **(right)** 3D example of the local curvature example, in which $f_1(x_1, x_2) = \exp(x_1) + \exp(x_2) - x_1 - x_2$ and $f_2(x_1, x_2) = f_1(-x_1, -x_2)$. This example highlights the need to toggle between functions according to their local curvature.

putational cost) to optimize a target objective (Forrester et al., 2007). In particular, the idea of using multiple closely-related tasks of varying levels of fidelity has seen adoption in settings where function evaluations are expensive, including bandits (Kandasamy et al., 2016b;a), Bayesian optimization (Kandasamy et al., 2017; Song et al., 2019; Wu et al., 2020; Takeno et al., 2020), and active learning (Yi et al., 2021; Li et al., 2020; 2022). The motivations behind the AMOO setting are clearly similar to those of proxy optimization and multi-fidelity optimization. However, our papers takes a pure optimization and gradient-descent perspective, which to our knowledge, is novel in the literature.

## 3. Aligned Multi Objective Optimization

Consider an unconstrained multi-objective optimization where $F : \mathbb{R}^n \to \mathbb{R}^m$ is a vector valued function, $F(\mathbf{x}) = (f_1(\mathbf{x}), f_2(\mathbf{x}), \ldots, f_m(\mathbf{x}))$, and all functions $\{f_i\}_{i \in [m]}$ are convex where $[m] := \{1, \ldots, m\}$. Without additional assumptions the components of $F(\mathbf{x})$ cannot be minimized simultaneously. To define a meaningful approach to optimize $F(\mathbf{x})$ one can study the Pareto front, or to properly define how to trade-off the objectives. In the AMOO setting we make the assumption the functions are aligned in a specific sense: we assume that the functions $\{f_i\}_{i \in [m]}$ share an optimal solution[1]. Namely, there exists a point $\mathbf{x}_\star$ that minimizes all functions in $F(\cdot)$ simultaneously,

$$\mathbf{x}_\star \in \arg \min_{\mathbf{x} \in \mathbb{R}^n} f_i(\mathbf{x}) \quad \forall i \in [m]. \tag{1}$$

With this assumption one may hope to get quantitative benefits from the multi objective feedback. How can Gradient Descent (GD) be improved when the functions are aligned?

A common algorithmic approach in the multi-objective setting is using a weight vector $\mathbf{w} \in \mathbb{R}^m$ that maps the vector $F(\mathbf{x})$ to a single objective $f_{\mathbf{w}}(\mathbf{x}) := \mathbf{w}^T F(\mathbf{x})$, and apply a gradient descent step on the weighted function (e.g., Sener and Koltun (2018); Yu et al. (2020); Liu et al. (2021)). Existing algorithms suggest alternatives for choosing $\mathbf{w}$ via different weight optimizers. We follow this paradigm and refer to it as `Weighted-GD` (see Algorithm 1).

We provide definitions of useful structures from convex optimization that will be used in this work.

**Definition 3.1** (Smoothness). $f(\mathbf{x}) : \mathbb{R}^n \to \mathbb{R}$ is called $\beta$-smooth if $\forall \mathbf{x}, \mathbf{y} \in \mathbb{R}^n$ the following holds:

$$f(\mathbf{y}) \leq f(\mathbf{x}) + \nabla f(\mathbf{x})^\top (\mathbf{y} - \mathbf{x}) + \frac{\beta}{2} \|\mathbf{x} - \mathbf{y}\|^2.$$

**Definition 3.2** (Self-concordant). $f(\mathbf{x}) : \mathbb{R}^n \to \mathbb{R}$ is called self-concordant with parameter $M_{\mathrm{f}} \geq 0$ if $\forall \mathbf{x}, \mathbf{y} \in \mathbb{R}^n$ the following holds:

$$\langle \nabla^3 f(\mathbf{x})[\mathbf{y}]\mathbf{y}, \mathbf{y} \rangle \preceq 2M_{\mathrm{f}} \|\mathbf{y}\|_{\mathbf{x}}^3,$$

where $\nabla^3 f(\mathbf{x})[\mathbf{y}] := \lim_{\alpha \to 0} \frac{1}{\alpha} \left( \frac{\nabla^2 f(\mathbf{x} + \alpha \mathbf{y}) - \nabla^2 f(\mathbf{x})}{\alpha} \right)$ is the directional derivative of the hessian in $\mathbf{y}$, and $\|\mathbf{y}\|_{\mathbf{x}}^2 := \|\mathbf{y}\|_{\nabla^2 f(\mathbf{x})}$.

Next, towards developing algorithmic intuition for the AMOO setting, we consider few examples in which alignment between different objective function may be useful.

---

[1]We also study an extension of AMOO where the functions can only be approximately simultaneously minimized. See Section 6.

*(i)* **The Specification Example.** Consider the case $F(\mathbf{x}) = (f_1(\mathbf{x}), f_2(\mathbf{x}))$, $\mathbf{x} \in \mathbb{R}^2$ where

$$f_1(\mathbf{x}) = (1 - \Delta)x_1^2 + \Delta x_2^2,$$
$$f_2(\mathbf{x}) = \Delta x_1^2 + (1 - \Delta)x_2^2,$$

for some small $\Delta \in [0, 0.1]$. It is clear that $F(\mathbf{x})$ can be simultaneously minimized in $\mathbf{x}_\star = (0, 0)$, hence, this is an AMOO setting. This example, as we demonstrate, illustrates an instance in which each individual function *does not specify the solution well*, but with proper weighting the optimal solution is well specified.

First, observe both $f_1$ and $f_2$ are $\Delta$-strongly convex and $O(1)$-smooth functions. Hence, GD with properly tuned learning rate, applied to either $f_1$ or $f_2$ converges with linear rate of $\Omega(\Delta)$. This rate can be dramatically improved by proper weighting of the functions. Let $f_{\mathbf{w}_U}$ be a function with equal weighting of both $f_1$ and $f_2$, namely, choosing $\mathbf{w}_U = (0.5, 0.5)$. We get $f_{\mathbf{w}_U}(\mathbf{x}) = 0.5x_1^2 + 0.5x_2^2$ which is $\Omega(1)$-strongly convex and $O(1)$-smooth. Hence, GD applied to $f_{\mathbf{w}_U}$ converges with linear rate of $\Omega(1)$–much faster than $O(\Delta)$ since $\Delta$ can be arbitrarily small.

*(ii)* **The Selection Example.** Consider the case $F(\mathbf{x}) = (f_1(\mathbf{x}), \ldots, f_m(\mathbf{x}))$, $\mathbf{x} \in \mathbb{R}^n$, where

$$\forall i \in [m-1]: f_i(\mathbf{x}) = (1 - \Delta)x_1^2 + \Delta \sum_{j=2}^{n} x_j^2,$$

$$f_m(\mathbf{x}) = \sum_{j=1}^{n} x_j^2,$$

and $\Delta \in [0, 0.5]$. The common minimizer of all functions is $\mathbf{x}_\star = \mathbf{0} \in \mathbb{R}^n$, and, hence, the objectives are aligned. Unlike the specification example, in the selection example, there is a single objective function among the $m$ objectives we should select to improve the convergence rate of GD. Further, in the selection example, choosing the uniform weight degrades the convergence rate.

Indeed, setting the weight vector to be uniform $\mathbf{w}_U = (1/m, \ldots, 1/m) \in \mathbb{R}^m$ leads to the function $f_{\mathbf{w}_U}(\mathbf{x}) = (2 - \Delta)/m \cdot x_1^2 + \sum_{j=2}^{n}(\Delta + 1)/m \cdot x_j^2$, which is $O(1/m)$-strongly convex. Hence, GD applied to $f_{\mathbf{w}_U}$ converges in a linear rate of $O(1/m)$. On the other hand, GD applied to $f_m$ converges with linear rate of $\Omega(1)$. Namely, setting the weight vector to be $(0, \ldots, 0, 1) \in \mathbb{R}^m$ improves upon taking the average when the number of objectives is large.

*(iii)* **Local Curvature Example.** Consider the case $F(x) = (f_1(x), f_2(x))$, $x \in \mathbb{R}$ where

$$f_1(x) = \exp(x) - x,$$
$$f_2(x) = \exp(-x) + x,$$

---

**Algorithm 1** `Weighted-GD`

---
**initialize:**
    Learning rate $\eta$, `Weight-Optimizer`
**while** $k = 1, 2, \ldots$ **do**
    $\mathbf{w}_k \leftarrow$ `Weight-Optimizer` $(\{f_i(\mathbf{x}_k)\}_{i=1}^m)$
    $\mathbf{g}_k \leftarrow \nabla f_{\mathbf{w}_k}(\mathbf{x}_k)$
    $\mathbf{x}_{k+1} = \mathbf{x}_k - \eta \mathbf{g}_k$
**end while**

---

namely, $f_2(x) = f_1(-x)$. Both functions are simultaneously minimized in $x = 0$. This example depicts a scenario in which different functions have a larger curvature in different segments of the parameter space; for $x > 0$, $f_1(x)$ has a larger curvature, and for $x < 0$ $f_2$ has a larger curvature.

For such a setting, it is natural to toggle between the two functions, namely to set the weight vector as $\mathbf{w}_1 = (1, 0)$ for $x > 0$ and as $\mathbf{w}_2 = (0, 1)$ for $x < 0$. This approach, intuitively, should result in a faster convergence to the optimal solution compared to applying GD to an individual function or the average function, since it effectively applies GD to a function which is uniformly more curved.

The three aforementioned examples highlight a key takeaway: the curvature of the functions has a vital role in improving convergence guarantees for GD in AMOO. Indeed, all examples provided arguments as of how to improve the convergence of GD based on curvature information. In next sections we formalize this intuition. We introduce quantities that characterize notions of best curvature, and develop new GD based algorithms that provably converge with rates that depend on these quantities.

## 4. The `CAMOO` Weight Optimizer

We start by introducing and analyzing the Curvature Aligned Multi Objective Optimizer (`CAMOO`). `CAMOO` (Algorithm 2) directly optimizes the curvature of the weighted function. Towards developing it, we define the global adaptive strong convexity parameter, $\mu_G$. Later we show that when the weighted loss is determined by `CAMOO` GD converges in a rate that depends on $\mu_G$.

We start by defining the optimal adaptive strong convexity over the class of weights:

**Definition 4.1** (Global Adaptive Strong Convexity $\mu_G$)**.** The global adaptive strong convexity parameter, $\mu_G \in \mathbb{R}_+$, is the largest value such that $\forall \mathbf{x} \in \mathbb{R}^n$ exists a weight vector $\mathbf{w} \in \Delta_m$ such that

$$\lambda_{\min}\left(\sum_{i=1}^{m} w_i \nabla^2 f_i(\mathbf{x})\right) \geq \mu_G. \tag{2}$$

For each $\mathbf{x} \in \mathbb{R}^n$, there may be a different weight vector that

solves $\arg \max \lambda_{\min} \left( \nabla^2 f_{\mathbf{w}}(\mathbf{x}) \right)$ and locally maximizes the curvature. The global adaptive strong convexity parameter $\mu_{\mathrm{G}}$ is the largest lower bound in $\mathbb{R}^n$. The specification and selection examples (Section 3) demonstrate $\mu_{\mathrm{G}}$ can be much larger than both the strong convexity parameter of the average function or of each individual function; for both $\mu_{\mathrm{G}} = O(1)$ whereas the alternatives may have arbitrarily small strongly convex parameter value. Further, the local curvature example highlights a case in which the optimal weight may have dependence on $\mathbf{x}$.

Additionally, this structural definition implies that there is a unique point $\mathbf{x}_\star$ that simultaneously minimizes the objectives. Due to this observation, in the following, we aim to design provable GD methods that converge to this optimal point $\mathbf{x}_\star$. The following result formalizes this by showing that under a weaker condition compared to $\mu_{\mathrm{G}} > 0$ there is a unique minimizer (see Appendix F.1 for a proof).

**Proposition 4.1** (Unique Optimal Solution). *Assume there exists $\mathbf{x}_\star \in \mathbb{R}^n$ that simultaneously minimizes $\{f_i\}_{i\in[m]}$, namely, solves Eq. (1). If $\max_{\mathbf{w}\in\Delta_m} \lambda_{\min} \left( \nabla^2 f_{\mathbf{w}}(\mathbf{x}_\star) \right) > 0$ then $\mathbf{x}_\star$ is unique.*

Definition 4.1 not only quantifies an optimal notion of curvature, but also directly results with the CAMOO algorithm. CAMOO sets the weights according to Eq. (2), namely, at the $k^{\mathrm{th}}$ iteration, it finds the weight vector for which $f_{\mathbf{w}}(\mathbf{x}_k)$ has the largest local curvature. Then, a gradient step is applied in the direction of $\nabla f_{\mathbf{w}}(\mathbf{x}_k)$ (see Algorithm 1). Indeed, CAMOO seems as a natural algorithm to apply in AMOO. Nevertheless, the analysis of CAMOO faces key challenges that make its analysis less trivial than what one may expect.

**Challenge *(i)*: $f_{\mathbf{w}_k}$ is not a strongly convex function.** One may hope that standard GD analysis for strongly convex and smooth functions can be applied. It is well known that if a function $f(\mathbf{x})$ is $\beta$ smooth and $\forall \mathbf{x} \in \mathbb{R}^n$, $\lambda_{\min} \left( \nabla^2 f(\mathbf{x}) \right) \geq \mu$ then GD converges with $\mu/\beta$ linear rate. Unfortunately, a careful examination of this argument shows it fails.

Even though $\lambda_{\min} \left( \nabla^2 f_{\mathbf{w}_k}(\mathbf{x}_k) \right) \geq \mu_{\mathrm{G}}$ at each iteration $k$ of CAMOO it does not imply that $f_{\mathbf{w}_k}$ is $\mu_{\mathrm{G}}$ strongly convex for a fixed $\mathbf{x}_k$. Namely, it does not necessarily hold that for all $\mathbf{x} \in \mathbb{R}^n$, $\lambda_{\min} \left( \nabla^2 f_{\mathbf{w}_k}(\mathbf{x}) \right) \geq \mu_{\mathrm{G}}$, but only pointwise at $\mathbf{x}_k$ (E.g., the local curvature example highlights this issue. See Appendix B.1 for details). This property emerges naturally in AMOO, yet such nuance is inherently impossible in single-objective optimization.

**Challenge *(ii)*: Weighted function is not necessarily convex.** A naive reduction may be to apply GD to the function $f_{\mathbf{w}_\star(\mathbf{x})}$ where $\mathbf{w}_\star(\mathbf{x}) \in \arg \max \lambda_{\min} \left( \sum_{i=1}^m w_i \nabla^2 f_i(\mathbf{x}) \right)$. Namely, to apply GD to a new weighted function that is determined by optimizing the curvature. Such an approach

---

**Algorithm 2** CAMOO

> **inputs:** $\{f_i(\mathbf{x}_k)\}_{i=1}^m$
> **initialize:** $w_{\min} = \mu_{\mathrm{G}} / (8m\beta)$
> Get Hessian matrices $\{\nabla^2 f_i(\mathbf{x}_k)\}_{i=1}^m$
> $\mathbf{w} \in \underset{\mathbf{w}\in\Delta_{m,w_{\min}}}{\arg \max} \; \lambda_{\min} \left( \sum_i w_i \nabla^2 f_i(\mathbf{x}_k) \right)$
> **return:** $\mathbf{w}$

---

turns out as flawed from theoretical perspective; the function $f_{\mathbf{w}_\star(\mathbf{x})} = \sum_{i\in[m]} w_{\star,i}(\mathbf{x}) f_i(\mathbf{x})$ is not necessarily convex nor smooth due to the dependence on a weight vector that has an $\mathbf{x}$ dependence (see Appendix B.2 for an example).

Next, we provide a positive result. When restricting the class of functions to the set of self-concordant and smooth functions (see Appendix C for formal definitions) we provide a convergence guarantee for Weighted-GD instantiated with CAMOO that depends on $\mu_{\mathrm{G}}$. Further, the result shows that close to the optimal solution the convergence has linear rate in $O(\mu_{\mathrm{G}}/\beta)$ (see Appendix D for proof details).

**Theorem 4.2** ($\mu_{\mathrm{G}}$ Convergence of CAMOO). *Suppose $\{f_i\}_{i\in[m]}$ are $\beta$ smooth, $M_{\mathrm{f}}$ self-concordant, share an optimal solution $\mathbf{x}_\star$ and that $\mu_{\mathrm{G}} > 0$. Let $k_0 := \left\lceil \frac{16\beta \left( \|\mathbf{x}_0 - \mathbf{x}_\star\| 3\sqrt{m}\beta M_{\mathrm{f}} - \sqrt{\mu_{\mathrm{G}}} \right)}{3\mu_{\mathrm{G}}^{3/2}} \right\rceil$, where $\|\cdot\|$ is the Euclidean-norm. Then, Weighted-GD instantiated with CAMOO weight-optimizer and $\eta = 1/2\beta$ converges with rate:*

$$\|\mathbf{x}_k - \mathbf{x}_\star\| \leq \begin{cases} \|\mathbf{x}_{k_0} - \mathbf{x}_\star\| \left( 1 - \frac{3\mu_{\mathrm{G}}}{8\beta} \right)^{(k-k_0)/2} & k \geq k_0 \\ \|\mathbf{x}_0 - \mathbf{x}_\star\| - k \frac{\mu_{\mathrm{G}}^{3/2}}{16\beta^2\sqrt{m}M_{\mathrm{f}}} & o.w. \end{cases}$$

Importantly, Theorem 4.2 holds without making strong convexity assumption on the individual functions, but only requires that the adaptive strong convexity parameter $\mu_{\mathrm{G}}$ to be positive, as, otherwise, the result is vacuous.

The proof follows few key observations. The self-concordance property, we find, implies a useful inequality that depends only on local curvature (see Appendix C):

$$f(\mathbf{y}) \geq f(\mathbf{x}) + \langle \nabla f(\mathbf{x}), \mathbf{y} - \mathbf{x} \rangle + \frac{c \|\mathbf{y} - \mathbf{x}\|_{\nabla^2 f(\mathbf{x})}^2}{1 + M_{\mathrm{f}} \|\mathbf{y} - \mathbf{x}\|_{\nabla^2 f(\mathbf{x})}}, \tag{3}$$

for some constant $c > 0$. This inequality share similarity with the more standard inequality used in analysis of GD convergence for strongly convex function (Boyd and Vandenberghe, 2004), however, unlike the former, it depends on local curvature. In order to satisfy the assumption the weighted function $f_{\mathbf{w}_k}$ is self-concordant we rely on the fact $\min_{i\in[m]} w_{k,i} > w_{\min}$ by design of CAMOO. Then, additional analysis leads to a recurrence relation of the residual

$r_k := \|\mathbf{x}_k - \mathbf{x}_\star\|^2$ with the form of

$$r_{k+1}^2 \leq r_k^2 - \alpha_1 r_k^2 / (1 + \alpha_2 r_k). \tag{4}$$

We provide a bound on this recurrence relation in Appendix C to arrive to the final result of Theorem 4.2.

### 4.1. Practical Implementation

We now describe a scalable approach for implementing CAMOO that we experiment with in next sections. Towards large scale application of CAMOO with modern deep learning architectures we approximate the Hessian matrices with their diagonal. Prior works used the diagonal Hessian approximation as pre-conditioner (Chapelle et al., 2011; Schaul et al., 2013; Yao et al., 2021; Liu et al., 2023b; Achituve et al., 2024). Notably, with this approximation the computational cost of CAMOO scales linearly with number of parameters in the Hessian calculation, instead of quadratically. The following result establishes that the value of optimal curvature, and, hence the convergence rate of Weighted-GD instantiated with CAMOO, degrades continuously with the quality of Hessian approximation (see Appendix F.2 for proof details).

**Proposition 4.2.** *Assume that for all $i \in [m]$ and $\mathbf{x} \in \mathbb{R}^n$ $\|\nabla^2 f_i(\mathbf{x}) - \mathrm{Diag}\left(\nabla^2 f_i(\mathbf{x})\right)\|_2 \leq \|\Delta\|$ where $\|\mathbf{A}\|_2$ is the spectral norm of $\mathbf{A} \in \mathbb{R}^{n \times n}$. Let $\mathbf{w}_\star \in \arg\max_{\mathbf{w} \in \Delta_m} \lambda_{\min}\left(\sum_i w_i \nabla^2 \mathrm{Diag}\left(f_i(\mathbf{x})\right)\right)$. Then, $\lambda_{\min}\left(\sum_i w_{\star,i} \nabla^2 f_i(\mathbf{x})\right) \geq \mu_{\mathrm{G}} - 2\|\Delta\|$.*

Next we provide high-level details of our implementation (also see Appendix A).

**Diagonal Hessian estimation via Hutchinson's Method.** We use the Hutchinson method (Hutchinson, 1989; Chapelle et al., 2011; Yao et al., 2021) which provides an estimate to the diagonal Hessian by averaging products of the Hessian with random vectors. Importantly, the computational cost of this method scales linearly with number of parameters.

**Maximizing the minimal eigenvalue.** Maximizing the minimal eigenvalue of symmetric matrices is a convex problem (Boyd and Vandenberghe (2004), Example 3.10) and can be solved via semidefinite programming. For diagonal matrices the problem can be cast as a simpler max-min bilinear problem, $\arg\max_{\mathbf{w} \in \Delta_m} \min_{\mathbf{q} \in \Delta_n} \mathbf{w}^\top \mathbf{A}\mathbf{q}$, where $n$ is the dimension of parameters, $\mathbf{A} \in \mathbb{R}^{m \times n}$ and its $i^{th}$ row is the diagonal Hessian of the $i^{th}$ objective, namely, $\forall i \in [m]$, $\mathbf{A}[i,:] = \mathrm{diag}(\nabla^2 f_i(\mathbf{x}))$.

This bilinear optimization problem is well studied (Rakhlin and Sridharan, 2013; Mertikopoulos et al., 2018; Daskalakis and Panageas, 2018). We implemented the PU method of Cen et al. (2021) which, loosely speaking, executes iterative updates via exponential gradient descent/ascent. PU

has a closed form update rule and its computational cost scales linearly with number of parameters.

## 5. The PAMOO Weight Optimizer

In previous section, we introduced the global adaptive strong convexity parameter, $\mu_{\mathrm{G}}$, the CAMOO weight optimizer that chooses the weight vector adaptively and showed it has asymptotic linear convergence guarantees that depend on $\mu_{\mathrm{G}}$. In this section we explore an additional adaptive mechanism for choosing the weight vector based on Polyak step-size design. We introduce the Polyak Aligned Multi-Objective Optimizer (PAMOO). Unlike CAMOO, it only requires information on the gradient, without requiring access to the Hessians. Interestingly, even though computationally much cheaper, PAMOO exhibits improved convergence rate compared to CAMOO.

PAMOO (Algorithm 3) generalizes the Polyak step-size design to AMOO. As such, it requires access to the optimal function values, $f_i(\mathbf{x}_\star)$ for all $i \in [m]$. This information may not be readily available in general. However, in modern machine learning applications this value is often zero (Loizou et al., 2021; Wang et al., 2023). Further, there are variations of Polyak step-size in which a the optimal value is estimated (Gower et al., 2021; Orvieto et al., 2022). We leave potential extensions of these to AMOO for future work. Compared to CAMOO, PAMOO only requires access to the gradients of the objectives, and does not assume access to the Hessians. Further, it only requires solving a simple convex quadratic optimization problem in dimension $\mathbb{R}^m$. This problem is simpler than a maximization of the smallest eigenvalue, required to solve by CAMOO.

We now define the local strong convexity parameter over a class of weights. As we later show, this parameter controls the convergence rate of PAMOO:

**Definition 5.1** (Local Strong Convexity $\mu_{\mathrm{L}}$). The local strong convexity parameter, $\mu_{\mathrm{L}} \in \mathbb{R}_+$, is the largest value such that exists a weight vector $\mathbf{w} \in \Delta_m$ such that

$$\lambda_{\min}\left(\sum_{i=1}^m w_i \nabla^2 f_i(\mathbf{x}_\star)\right) \geq \mu_{\mathrm{L}}, \tag{5}$$

where $\mathbf{x}_\star$ simultaneously minimizes $\{f_i\}_{i \in [m]}$.

Notice that Proposition 4.1 implies that $\mathbf{x}_\star$ is necessarily unique, and, hence $\mu_{\mathrm{L}}$ is unique and well defined. Further, unlike the global adaptive strong convexity parameter, the local strong convexity parameter only depends on the curvature at $\mathbf{x}_\star$, namely, at the optimal solution. From Definition 4.1 and Definition 5.1 we directly get that $\mu_{\mathrm{L}} \geq \mu_{\mathrm{G}}$.

The PAMOO algorithm is inspired by the Polyak step-size design (Polyak, 1987; Hazan and Kakade, 2019) for choosing the learning rate in a parameter-free way. To provide with

**Algorithm 3** `PAMOO`

---
1: **inputs:** $\{f_i(\mathbf{x}_k)\}_{i=1}^m$
2: $\mathbf{w} \in \arg\max_{\mathbf{w} \in \mathbb{R}_+^m} 2\mathbf{w}^\top \Delta_{\mathbf{x}} - \mathbf{w}^\top \mathbf{J}_{\mathbf{x}}^\top \mathbf{J}_{\mathbf{x}} \mathbf{w}$
3: $\Delta_{\mathbf{x}} := [\Delta_{\mathbf{x},1} \ldots \Delta_{\mathbf{x},m}], \Delta_{\mathbf{x},i} := f_i(\mathbf{x}_k) - f_i(\mathbf{x}_\star)$
4: $\mathbf{J}_{\mathbf{x}} := [\nabla f_1(\mathbf{x}) \ldots \nabla f_m(\mathbf{x})] \in \mathbb{R}^{n \times m}$
5: **return: w**

---

intuition for our derivation, consider the GD update rule in a single objective optimization problem, $\mathbf{x}_{k+1} = \mathbf{x}_k - \eta_k \mathbf{g}_k$. To derive the Polyak step-size design, observe that by convexity and the GD update rule we have that

$$\|\mathbf{x}_{k+1} - \mathbf{x}_\star\|^2 \tag{6}$$
$$\leq \|\mathbf{x}_k - \mathbf{x}_\star\|^2 - 2\eta_k \left(f(\mathbf{x}_k) - f(\mathbf{x}_\star)\right) + \eta_k^2 \|\nabla f(\mathbf{x}_k)\|^2.$$

Minimizing the upper bound on the decrease with respect to $\eta_k$ leads to $\eta_k = \left(f(\mathbf{x}_k) - f(\mathbf{x}_\star)\right) / \|\nabla f(\mathbf{x}_k)\|^2$, which is the Polyak step-size design choice.

Building on this derivation we develop the `PAMOO` weight optimizer (see Algorithm 3). As we now show, interestingly, its convergence rate has the same functional form as `CAMOO`, while depending on the local strong convexity parameter $\mu_{\mathrm{L}}$ instead in $\mu_{\mathrm{G}}$. Hence, `PAMOO` has an improved upper bound on its convergence rate compared to `CAMOO` (see Appendix D for proof details).

**Theorem 5.2** ($\mu_{\mathrm{L}}$ Convergence of `PAMOO`)**.** *Suppose* $\{f_i\}_{i \in [m]}$ *are* $\beta$ *smooth,* $M_{\mathrm{f}}$ *self-concordant, share an optimal solution* $\mathbf{x}_\star$ *and* $\mu_{\mathrm{L}} > 0$*. Let* $k_0 := \left\lceil \frac{64\beta\left(\|\mathbf{x}_0 - \mathbf{x}_\star\| 3\sqrt{m}\beta M_{\mathrm{f}} - \sqrt{\mu_{\mathrm{L}}}\right)}{3\mu_{\mathrm{L}}^{3/2}} \right\rceil$*, where* $\|\cdot\|$ *is the Euclidean-norm. Then,* `Weighted-GD` *instantiated with* `PAMOO` *weight-optimizer and* $\eta = 1$ *converges with rate:*

$$\|\mathbf{x}_k - \mathbf{x}_\star\| \leq \begin{cases} \|\mathbf{x}_{k_0} - \mathbf{x}_\star\| \left(1 - \frac{3\mu_{\mathrm{L}}}{32\beta}\right)^{(k-k_0)/2} & k \geq k_0 \\ \|\mathbf{x}_0 - \mathbf{x}_\star\| - k \frac{\mu_{\mathrm{L}}^{3/2}}{64\beta^2\sqrt{m}M_{\mathrm{f}}} & o.w. \end{cases}$$

This result is established by generalizing the Polyak step-size method analysis (see Eq. (6)) while using a key observation. In the analysis, we upper bound the residual $\|\mathbf{x}_k - \mathbf{x}_\star\|^2$ by a quantity that depends on the curvature of the optimal weight vector at $\mathbf{x}_\star$, which is lower bounded by $\mu_{\mathrm{L}}$, by definition. This is valid since we can replace $\mathbf{w}_k$ with an alternative weight vector – only used in the analysis – since $\mathbf{w}_k$ is an optimal solution of $\max_{\mathbf{w} \in \mathbb{R}_+^m} 2\mathbf{w}^\top \Delta_{\mathbf{x}} - \mathbf{w}^\top \mathbf{J}_{\mathbf{x}}^\top \mathbf{J}_{\mathbf{x}} \mathbf{w}$. This flexibility allows us to upper bound expressions that depend on $\mathbf{w}_k$ by any nonnegative weight vector $\mathbf{w} \in \mathbb{R}_+^m$. Furthermore, we use similar tools as were developed in the analysis of `CAMOO`: the property of self-concordant functions (Eq. (3)) and the recurrence relation bound (Eq. (4)).

## 5.1. Practical Implementation

`PAMOO` can be implemented in a straightforward and scalable way. It requires access to the Jacobian matrix, which can be readily calculated by accessing the gradients. Calculating the matrix $\mathbf{J}_{\mathbf{x}}^\top \mathbf{J}_{\mathbf{x}} \in \mathbb{R}^{m \times m}$ has a computational cost of $O(nm^2)$, where $n$ is the dimension of the parameter space and $m$ is the number of objectives, and can be parallelized. Lastly, it requires solving a quadratic convex optimization problem in $\mathbb{R}^m$ where $m$ is expected to be of the order of $\sim 10$. This can be done efficiently with different convex optimization algorithms, e.g., projected GD. In practice, we initialize the weight vector $\mathbf{w}$ using the weight vector of the previous iterate. Hence, an approximate optimal solution is found within a few projected gradient descent iterations. `PAMOO` as a potential advantage over `CAMOO` due to its scalability and its simple implementation. Lastly, generalizing it to methods in which the optimal value is estimated instead of being given is left for future work (Orvieto et al., 2022; Gower et al., 2021).

## 6. $\epsilon$-AAMOO: Robustness to Alignment Assumption

We have analyzed `CAMOO` and `PAMOO` assuming perfectly aligned objectives (Eq. (1)). However, in practice, objectives may be 'similar' rather than perfectly aligned. Next, we extend AMOO to address this more realistic scenario and assume that the alignment assumption is approximately correct. We show that both algorithms are robust to such an approximation and remain effective under these conditions.

Instead of assuming the objectives are perfectly aligned, we consider the $\epsilon$-Approximate AMOO (AAMOO) framework, in which there exists a near-optimal solution with respect to all objectives. Let $\mathcal{C}_\epsilon$ be the set of $\epsilon$-approximate solutions:

$$\mathcal{C}_\epsilon = \left\{\mathbf{x} \in \mathbb{R}^n \mid f_i(\mathbf{x}) - f_i(\mathbf{x}_\star^i) \leq \epsilon \ \forall i \in [m]\right\}, \tag{7}$$

where $\mathbf{x}_\star^i \in \arg\min_{\mathbf{x} \in \mathbb{R}^n} f_i(\mathbf{x})$. In $\epsilon$-AAMOO setting we assume that $\mathcal{C}_\epsilon$ is not the empty set. This corresponds to a case in which exists a point that is a near-optimal solution for all objectives and can be understood as a natural generalization of the stricter AMOO setting in which $\epsilon = 0$.

The main result of this section shows that for both `CAMOO` and `PAMOO` the distance between $\mathbf{x}_k$ and an $\epsilon$-approximate solution $\mathbf{x}_\star^\epsilon \in \mathcal{C}_\epsilon$, converges to $\epsilon_{\mathrm{app}}$. Further, $\epsilon_{\mathrm{app}}$ is a polynomial function of $\epsilon$, structural quantities of the problem, and vanishes as $\epsilon \to 0$. This provides an approximate convergence guarantee of both algorithms. For `CAMOO`, the result depends on the $\mu_{\mathrm{G}}$ curvature, as in the AMOO setting. For `PAMOO` the convergence depends on the best curvature within the set $\mathcal{C}_\epsilon$ defined as follows:

**Definition 6.1** ($\epsilon$-Local Strong Convexity $\mu_{\mathrm{L}}^\epsilon$)**.** The $\epsilon$-local strong convexity, $\mu_{\mathrm{L}}^\epsilon \in \mathbb{R}_+$, is the largest value such that

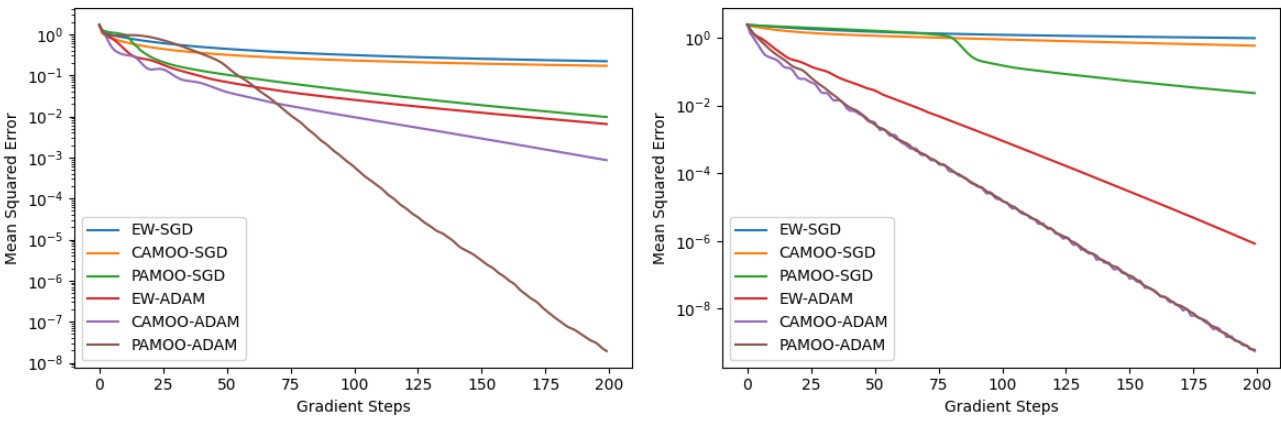

Figure 2: MSE (see Eq. (9)) versus gradient steps. **(left)** local curvature example instance, **(right)** selection example instance
.

$\exists \mathbf{x} \in \mathcal{C}_\epsilon$ exists a weight vector $\mathbf{w} \in \Delta_m$ such that

$$\lambda_{\min}\left(\sum_{i=1}^m w_i \nabla^2 f_i(\mathbf{x})\right) \geq \mu_{\mathrm{L}}^\epsilon. \tag{8}$$

Let $\mu_\star(\mathbf{x}) = \max_{\mathbf{w} \in \Delta_m} \lambda_{\min}\left(\sum_{i=1}^m w_i \nabla^2 f_i(\mathbf{x})\right)$ be the largest curvature at point $\mathbf{x}$. Then, $\mu_{\mathrm{L}}^\epsilon$ is defined as the *maximal* largest curvature in the set of near optimal solutions $\mathcal{C}_\epsilon$, namely, $\max_{\mathbf{x} \in \mathcal{C}_\epsilon} \mu_\star(\mathbf{x})$. Unlike $\mu_{\mathrm{G}}$ that depends on the worst case curvature at all points, $\mu_{\mathrm{L}}^\epsilon$ depends on the best-case curvature in $\mathcal{C}_\epsilon$. PAMOO convergence depends on this quantity, which also satisfies $\mu_{\mathrm{L}}^\epsilon \geq \mu_{\mathrm{G}}$ for any $\epsilon$.

We now provide an informal description of an approximate convergence guarantee for both CAMOO and PAMOO (see Appendix E for the formal theorems and proofs):

**Theorem 6.2** ((Informal) Approximate Convergence in $\epsilon$-AAMOO)**.** *Suppose $\{f_i\}_{i \in [m]}$ are $\beta$ smooth and $M_{\mathrm{f}}$ self-concordant. Let $\mu_{\mathcal{A}}$ be $\mu_{\mathrm{G}}$ and $\mu_{\mathrm{L}}^\epsilon$ for CAMOO and PAMOO, respectively. Assume that $\epsilon$-AAMOO holds and that $\epsilon \leq \mathrm{poly}\left(\mu_{\mathcal{A}}, 1/\beta, 1/m, M_{\mathrm{f}}\right)$. Then, exists $\mathbf{x}_\star^\epsilon \in \mathcal{C}_\epsilon$ such that for both CAMOO and PAMOO iterates satisfy*

$$\|\mathbf{x}_k - \mathbf{x}_\star^\epsilon\| \leq \begin{cases} f_1 (1 - c\frac{\mu_{\mathcal{A}}}{\beta})^{(k-k_0)/2} + f_2 \epsilon^{1/4} & k \geq k_0 \\ \|\mathbf{x}_0 - \mathbf{x}_\star^\epsilon\| - f_3 k & o.w. \end{cases}$$

*Where $c \in (0,1)$, $f_1 = \|\mathbf{x}_{k_0} - \mathbf{x}_\star^\epsilon\|$ and $f_2, f_3$ and $k_0$ are polynomial functions of $\beta, \mu_{\mathcal{A}}, m$ and $M_{\mathrm{f}}$.*

Namely, the performance of CAMOO and PAMOO degrades continuously with $\epsilon$, as the alignment assumption is violated. This holds without modification to the algorithms.

## 7. Toy Experiment

We implemented CAMOO, PAMOO and compared them to a weighting mechanism that uses equal weights on the objectives (EW)[2]. We tested these three algorithms as the `Weight-Optimizers` in `Weighted-GD` (see Algorithm 1). We experimented with SGD and ADAM as the optimizers of the weighted loss. In the learning problem we consider one network is required to match the outputs of a second fixed network. We denote the fixed network with parameters $\theta_\star$ as $h_{\theta_\star} : \mathbb{R}^{d_i} \to \mathbb{R}^{d_o}$ and the second network with parameters $\theta$ as $h_\theta : \mathbb{R}^{d_i} \to \mathbb{R}^{d_o}$. Both are 2-layer neural networks with relu-activation and 512 hidden units.

We draw data from a uniform distribution $\mathcal{D} = \{\mathbf{x}_i\}_i$ where $\mathbf{x}_i \in \mathrm{Uniform}([-1,1]^{d_i})$. We consider three loss functions:

$$f_i(\theta) = \frac{1}{|\mathcal{D}|}\sum_{\mathbf{x} \in \mathcal{D}}\left((h_\theta(\mathbf{x}) - h_{\theta_\star}(\mathbf{x}))^\top \mathbf{H}_i (h_\theta(\mathbf{x}) - h_{\theta_\star}(\mathbf{x}))\right)^{\alpha_i},$$

for all $i \in [3]$, where $\mathbf{H}_i \in \mathbb{R}^{d_o \times d_o}$ is a positive definite matrix and $\alpha_i \geq 1$. All loss functions are minimized when $h_\theta(\mathbf{x}) = h_{\theta_\star}(\mathbf{x})$, and, hence, it is an instance of AMOO. We investigated two instances. First, a selection example in which $\alpha_i = 1$, and $\mathbf{H}_i = \mathrm{diag}(1, 0.01^i, \cdots, 0.01^i)$ for $i \in \{0,1,2\}$. There we expect the algorithms to adapt to the first loss function, $f_1$. Second, a local curvature example where $\mathbf{H}_i = \mathbf{I}$, and $\alpha_i \in \{1, 1.5, 2\}$. For such a choice $f_1$ has larger curvature for large losses whereas $f_2$ and $f_3$ have larger curvature for small losses.

We track the performance by measuring the mean-squared error between the networks:

$$\mathrm{MSE} = \frac{1}{|\mathcal{D}|}\sum_{\mathbf{x} \in \mathcal{D}}\|(h_\theta(\mathbf{x}) - h_{\theta_\star}(\mathbf{x}))\|. \tag{9}$$

Figure 2 shows the convergence plots of our experiments. These highlight the potential of using GD algorithms that designed for AMOO: both CAMOO and especially PAMOO

---

[2]https://github.com/facebookresearch/AlignedMultiObjectiveOptimization

show an improved convergence rate. We provide additional experimental results and details in Appendix A. The results show that `CAMOO` modifies the weights, as expected, by adapting them to the local curvature; i.e., it changes the weights gradually during the optimization phase.

## 8. Conclusions

In this work, we introduced the AMOO framework to study how aligned or approximately aligned multi-objective feedback can improve gradient descent convergence. We designed the `CAMOO` and `PAMOO` algorithms, which adaptively weight objectives and offer provably improved convergence guarantees. Future research directions include determining optimal rates for AMOO and conducting comprehensive empirical studies in different domains. Additionally, in this work, we have not explored a stochastic or non-convex optimization frameworks of AMOO, which we believe is of interest for future work. We conjecture that algorithmic advancements in AMOO will improve our ability to scale learning algorithms to handle large number of related tasks efficiently with minimal hyper-parameter tuning; a goal much needed in modern machine learning practice.

## Impact Statement

This paper presents work whose goal is to advance the field of Machine Learning and optimization. There are many potential societal consequences of our work, none which we feel must be specifically highlighted here.

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

# A. Additional Experimental Details

We share an IPython notebook in which we implement our algorithms and was used to generate all plots accompanies this submission.

**Dataset.** We sample 200 points from an independent uniform distribution $\mathbf{x}_i \in \text{Uniform}([-1,1]^{20})$. We generate target data from a randomly initialized network $t(x_i) = h_{\theta_\star}(x_i) + 10$. The target dimension is 7.

**Network architecture.** We choose the ground truth network and target network to have the same architecture. Both are 2 layer neural networks with 512 hidden dimensions and ReLu activation. The neural network outputs a vector in dimension 7.

**Training.** For both problems, we set the learning rate of SGD to 0.0005 and ADAM to 0.005, we use the same learning rates for PAMOO. For CAMOO we multiply the learning rate by the number of loss functions, 3. While equal weighting sets all weights to 1, CAMOO is constrained to set the sum of all weights to one, our adjustment accounts for this normalization.

**General parameters for CAMOO.** We set the number of samples for the Hutchinson method to be $N_{\text{Hutch}} = 10$. Namley, we estimate the Hessian matrices by averaging $N_{\text{Hutch}} = 10$ estimates obtained from the Hutchnison method. Further, at each training step we perform a single update of the weights based on the PU update rule of Cen et al. (2021) to solve the max-min Bilinear optimization problem (see Section 4.1). We use their primal-dual algorithm, and choose the learning rate as they specified $1/(2\max_{i,j} |\mathbf{A}_{ij}| + \tau_{\text{CAMOO}})$ where $\mathbf{A}$ is the matrix in the Bilinear optimization problem and $\tau_{\text{CAMOO}} = 0.01$ is a regularization parameter we choose. Additionally, we set the number of iterations of the primal-dual algorithm to be 100 per-step. We did not choose either parameter with great care, exploring best settings is part of future explorations.

**General parameters for PAMOO.** We solved the constraint convex optimization problem in PAMOO (see Algorithm 3) via the projected GD algorithm, where the projection on $\mathbb{R}^m_+$ is done by clipping negative values to $10^{-6}$. We set the learning rate to be $3e^{-3}$, and added a small regularization $\mathbf{J}_\mathbf{x}^\top \mathbf{J}_\mathbf{x} \to \mathbf{J}_\mathbf{x}^\top \mathbf{J}_\mathbf{x} + \tau_{\text{PAMOO}}\mathbf{I}$ to avoid exploding weights, where $\tau_{\text{PAMOO}} = 1e^{-4}$.

**Additional Plots of Experiments in Section 7**. For completeness, we present the three loss functions of the objectives $\{f_i\}_{i\in[3]}$ and the weighted loss as a function of GD iterates. Figure 3 depicts the losses for the local curvature example instance, and Figure 4 depicts the losses for the selection example instance. Unlike EW, CAMOO and PAMOO adaptively modify the weight vector and adjust it to current parameters.

We measured the weight vector as a function of GD iterates and present the results in Figure 9.

- **Local curvature example instance.** For the local curvature example instance we expect to see a switching behavior in CAMOO (see Figure 5). Namely, when the loss is large, the weight vector of CAMOO should assign most of its weight to the quartic loss function, $f_3$, since it has the largest curvature for large deviations from optimality. For small deviations, we expect CAMOO to assign weights to the quadratic loss, $f_1$.

- **Selection example instance.** For the selection example instance we expect CAMOO to set the weights as a 1-hot vector on $f_1$, since its the quadratic function with largest curvature across all dimensions. Figure 7 exemplifies this.

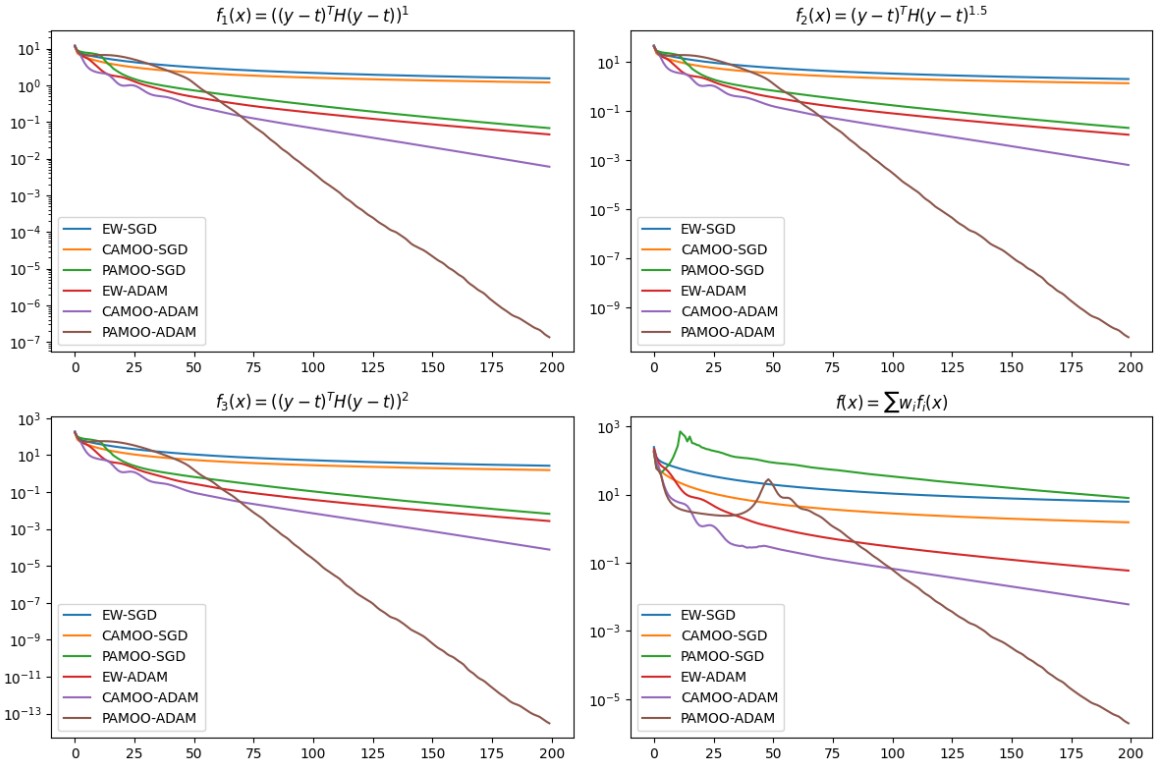

Figure 3: Local curvature example, all loss functions.

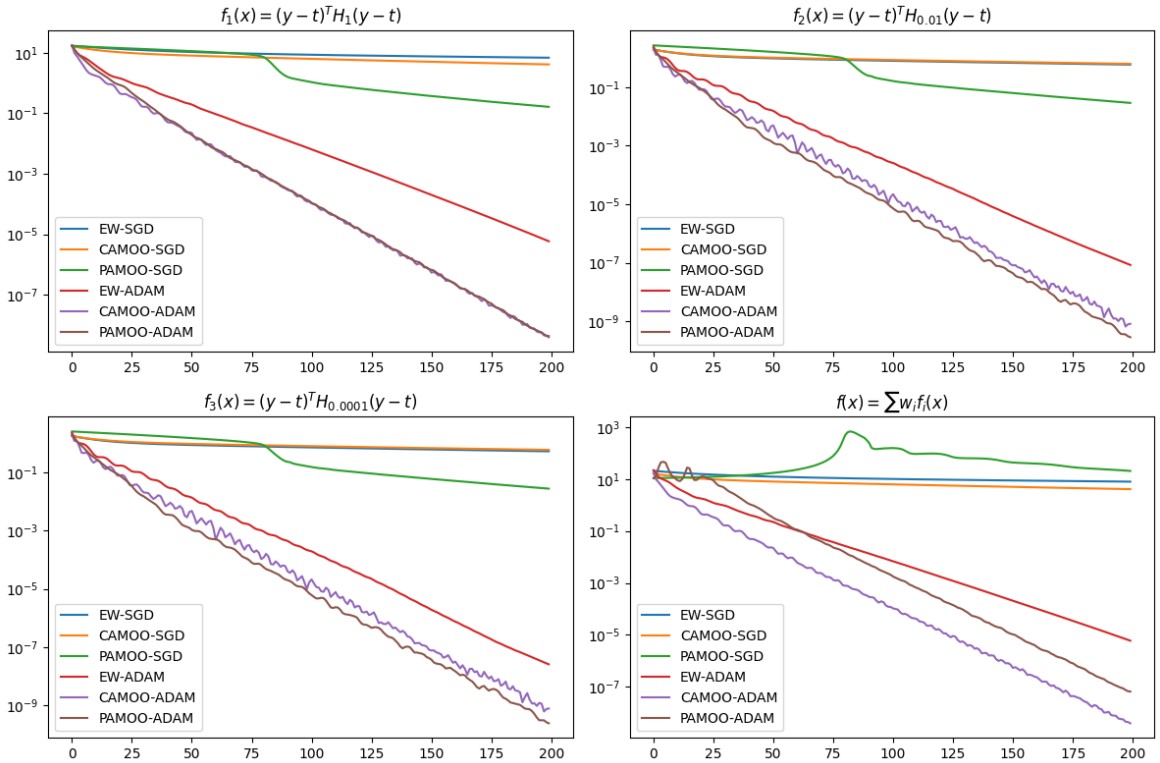

Figure 4: Selection example, all loss functions.

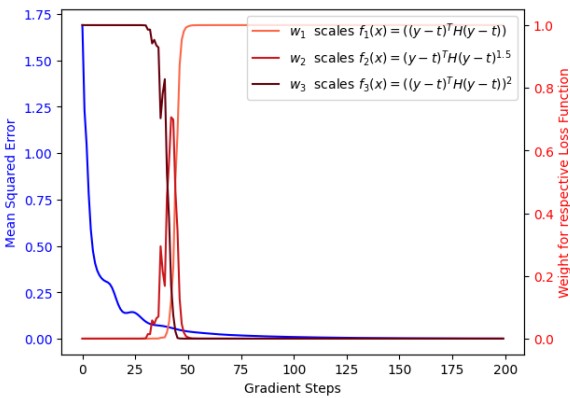

Figure 5: Local curvature example, `CAMOO`. The weights flip when the curvature of the loss function changes.

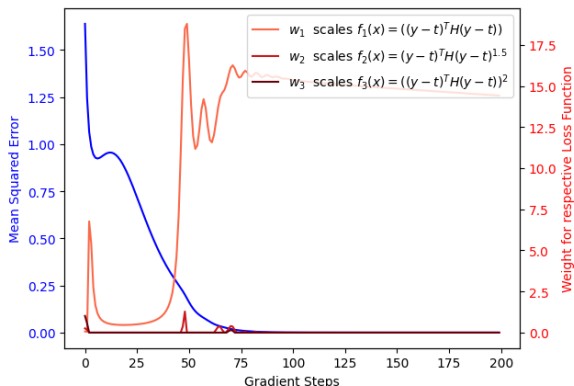

Figure 6: Local curvature example, `PAMOO`.

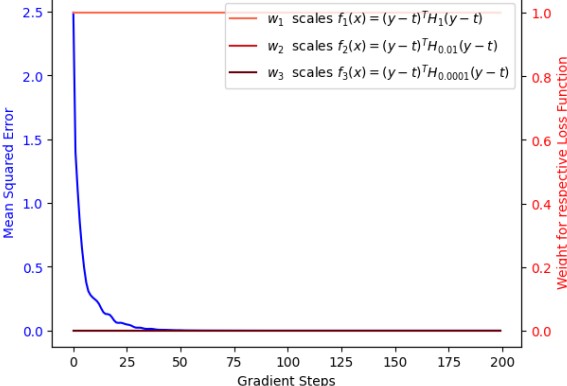

Figure 7: Selection example, `CAMOO`.

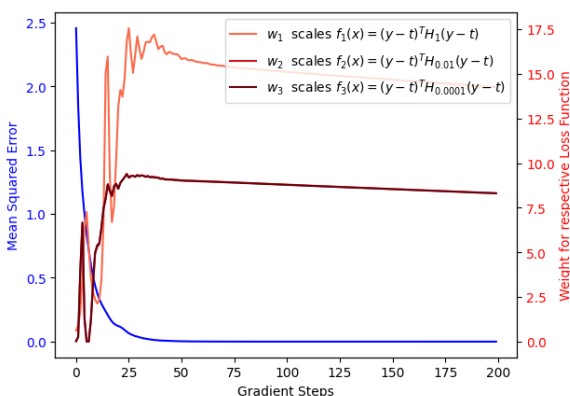

Figure 8: Selection example, `PAMOO`.

Figure 9: Weight vector evolution versus GD iterates.

## A.1. Toy Experiments: $\epsilon$ Approximate AMOO

We include additional experiment for the approximate AMOO setting, in which there exists a near-optimal solution for all objectives. We consider 3 loss functions of the local curvature example and add an approximation error as follows:

$$f_i(\theta) = \frac{1}{|\mathcal{D}|} \sum_{\mathbf{x} \in \mathcal{D}} \big( (h_\theta(\mathbf{x}) - h_{\theta_\star}(\mathbf{x}) + \epsilon_i)^\top \mathbf{H}_i (h_\theta(\mathbf{x}) - h_{\theta_\star}(\mathbf{x})) + \epsilon_i \big)^{\alpha_i},$$

where we set $H_i = I$ and $\epsilon_i \in \{0, \epsilon, -\epsilon\}$ where $\epsilon = 0.01$.

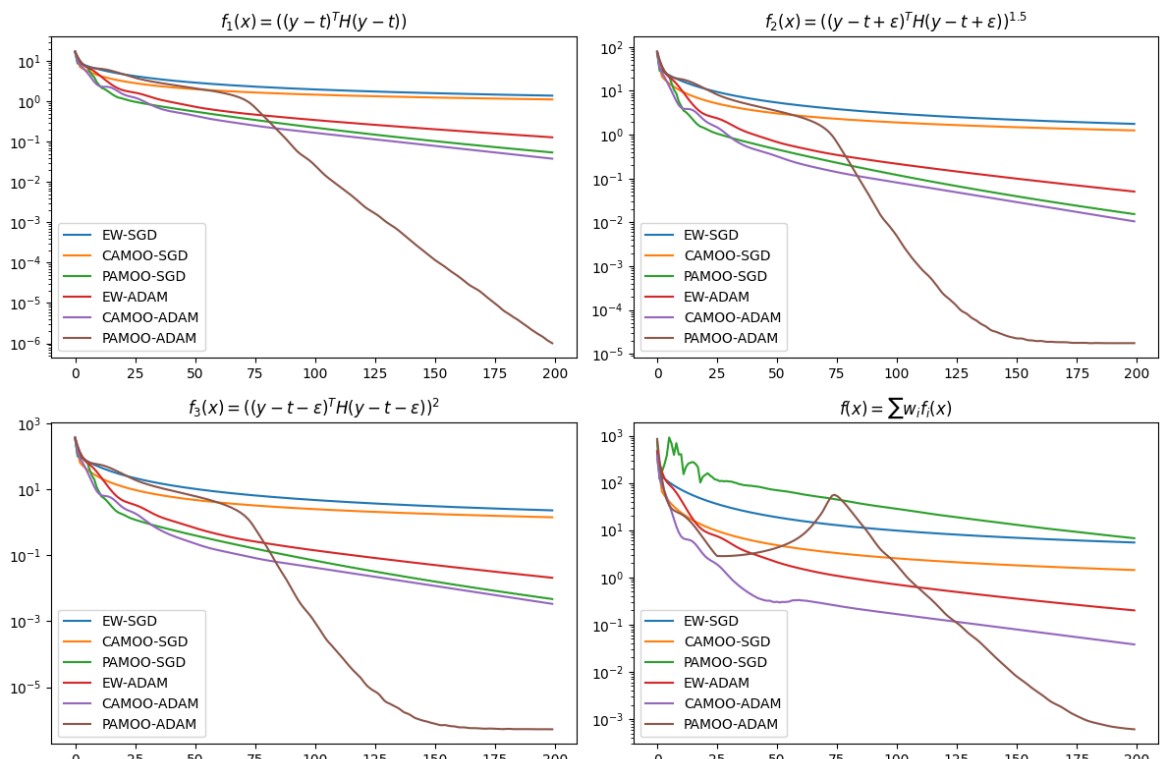

Figure 10: Local curvature example for $\epsilon$-approximate AMOO, all loss functions.

### A.2. Toy Experiments: Reducing Computational Complexity of `PAMOO`

Both `CAMOO` and `PAMOO` solve convex optimization problems in their inner loop to update the weight vector $\mathbf{w}$. In our implementations we use simple first-order algorithms that apply $N$ gradient-descent updates as detailed in Section 4 and Section 5. The results in the main paper depicts the performance of `CAMOO` and `PAMOO` when $N = 100$ and $N = 1000$, respectively. For these values, both `CAMOO` and `PAMOO` are significantly more costly compared to the `EW` algorithm.

Here we experiment we cheaper variations of both `CAMOO` and `PAMOO` in which we set $N = 10$, and, hence, reduce the number of updates to the weight vector. Table 2 shows the improvement in running time of `CAMOO` and `PAMOO` as well as comparing their running time to `EW`. As observed, the running time of `PAMOO` can be significantly improved whereas the running time of `CAMOO` is worse than `EW`. This is caused by the need to estimate Hessian information which can be costly without careful optimization. Figure 11 depicts the performance of `CAMOO` and `PAMOO` when $N = 10$. The performance of both algorithms degraded on the expense of improved running time. This shows that as the running time is improved algorithm designer may expect to experience performance degradation.

| Algorithm | Runtime (s) |
|---|---|
| *EW Algorithms* | |
| EW-SGD | 0.408452 |
| EW-ADAM | 0.480058 |
| *CAMOO and PAMOO* | |
| PAMOO-SGD | 8.654938 |
| CAMOO-SGD | 6.695782 |
| PAMOO-ADAM | 8.435489 |
| CAMOO-ADAM | 7.466246 |
| *CAMOO and PAMOO with $N = 10$* | |
| PAMOO-SGD | 0.715199 |
| CAMOO-SGD | 3.674786 |
| CAMOO-ADAM | 3.598157 |
| PAMOO-ADAM | 0.726002 |

Table 2: Runtime in seconds.

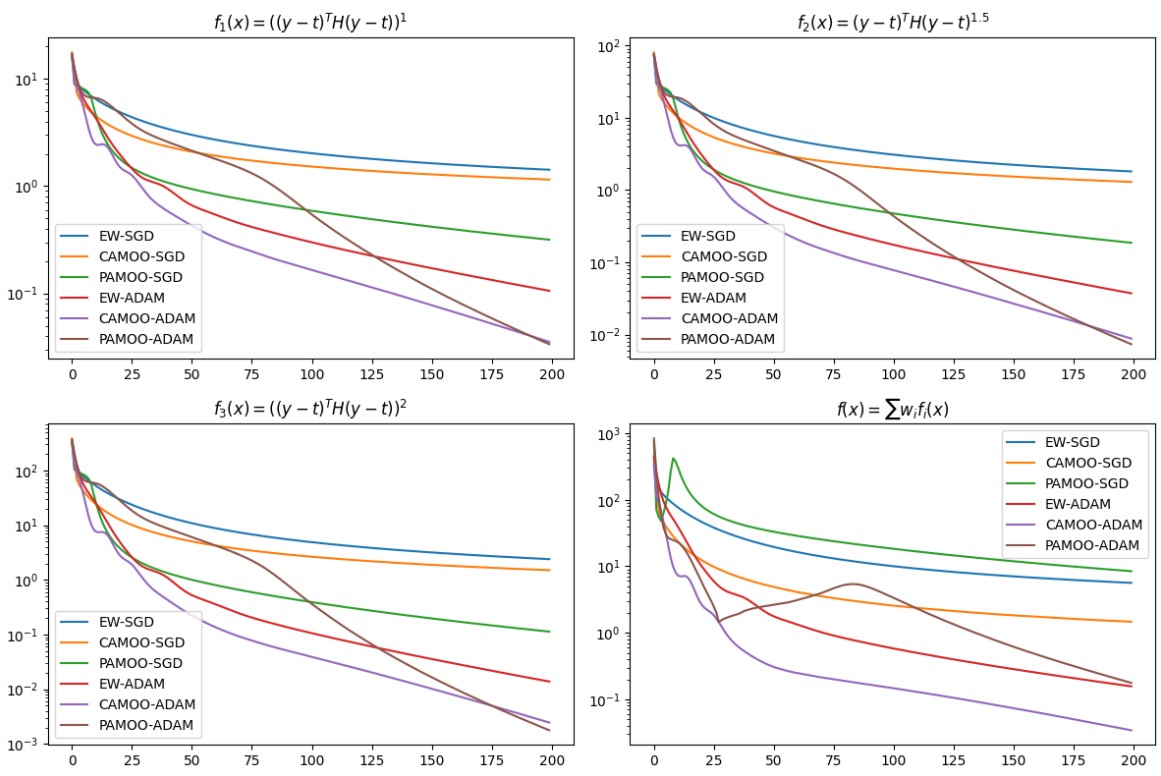

Figure 11: Local curvature example with reduced computational complexity ($N = 10$).

# B. Challenges in Analyzing `CAMOO`

## B.1. $f_{\mathbf{w}_k}$ is not a Strongly Convex Function

We provide an example that shows that for the AMOO setting, when $\mu_{\mathrm{G}} > 0$, the fact that $\lambda_{\min}\left(\nabla^2 f_{\mathbf{w}_k}(\mathbf{x}_k)\right) \geq \mu_{\mathrm{G}}$ throughout the iterates of `CAMOO` does not imply that each $f_{\mathbf{w}_k}$ is $\mu_{\mathrm{G}}$-strongly convex.

The counter example is the local curvature example (see Section 3), namely,

$$f_1(x) = \exp(x) - x, f_2(x) = \exp(-x) + x,$$

and $x \in \mathbb{R}$. The minimum point of both $f_1$ and $f_2$ is at $x = 0$. The Hessians of the functions are:

$$\nabla^2 f_1(x) = \exp(x), \nabla^2 f_2(x) = \exp(-x). \tag{10}$$

We see that $\nabla^2 f_1(x) > 1 > \nabla^2 f_2(x)$ for $x > 0$, $\nabla^2 f_1(x) < 1 < \nabla^2 f_2(x)$ for $x <= 0$, and $\nabla^2 f_1(0) = \nabla^2 f_2(0) = 1$.

This implies that $\mu_{\mathrm{G}} = 1$, and `CAMOO` will set the weight vector as $\mathbf{w}_+ = (1, 0)$ for $x > 0$ and $\mathbf{w}_- = (0, 1)$ for $x \leq 0$. However, it is readily observed that neither $f_{\mathbf{w}_-}$ or $f_{\mathbf{w}_+}$ are 1-strongly convex functions. The smallest value of the individual Hessians is zero since

$$\inf_{x \in \mathbb{R}} \nabla^2 f_{\mathbf{w}_-}(x) = \inf_{x \in \mathbb{R}} \exp(x) = 0$$
$$\inf_{x \in \mathbb{R}} \nabla^2 f_{\mathbf{w}_+}(x) = \inf_{x \in \mathbb{R}} \exp(-x) = 0,$$

by Eq. (10). Hence, the functions $f_{\mathbf{w}_k}$ produced in the iterates of `CAMOO` are not strongly convex.

## B.2. Weighted function is not necessarily convex

We provide a simple counter example that shows failure of a naive reduction in which we construct $f_{\mathbf{w}_\star(\mathbf{x})}(\mathbf{x}) = \sum_i w_{\star,i}(\mathbf{x}) f_i(\mathbf{x})$, where $\mathbf{w}_\star(\mathbf{x})$ optimizes the curvature at each point via,

$$\mathbf{w}_\star(\mathbf{x}) \in \arg\max_{\mathbf{w} \in \Delta_m} \lambda_{\min} \left( \sum_{i \in [m]} w_i \nabla^2 f_i(\mathbf{x}_k) \right)$$

and apply gradient descent to the function $f_{\mathbf{w}_\star(\mathbf{x})}$. The reason such approach is problematic is the fact that $f_{\mathbf{w}_\star(\mathbf{x})}$ may no longer be a convex function. This is a result of the extra dependence on $\mathbf{x}$ of $\mathbf{w}_\star(\mathbf{x})$.

A counter example can be established for a simple scenario in which

$$f_1(x) = x^2, f_2(x) = x^2 + c,$$

where $x \in \mathbb{R}$, for some $c \neq 0$. The two functions are convex, and have a minimizer at $x = 0$. However, since the Hessians of the functions are equal, $\nabla^2 f_1(x) = \nabla^2 f_2(x)$, the solution of the optimization problem

$$\mathbf{w}_\star(x) \in \arg\max_{\mathbf{w} \in \Delta_m} \lambda_{\min} \left( \sum_{i \in \{1,2\}} w_i \nabla^2 f_i(x) \right), \tag{11}$$

is arbitrary. Namely, each point on the simplex is a solution of this optimization problem. For example, choosing

$$\mathbf{w}_\star(x) = (\sin^2(x), \cos^2(x)),$$

is a solution of Eq. (11). With this, the function $f_{\mathbf{w}_\star(x)}$ takes the form of

$$f_{\mathbf{w}_\star(x)} = x^2 + c \times \cos^2(x),$$

which is not a convex function if, for example, $|c| > 1$.

Additionally, choosing $\mathbf{w}_\star(x)$ as non-smooth function, for example

$$\mathbf{w}_\star(x) = \begin{cases} (0,1) & x > 0 \\ (1,0) & x \leq 0, \end{cases}$$

results with a non-smooth function $f_{\mathbf{w}_\star(x)}$. This highlights that differentiating the function $f_{\mathbf{w}_\star(x)}$ is flawed from a theoretical perspective: $f_{\mathbf{w}_\star(x)}$ is not necessarily convex nor smooth.

## C. Preliminaries and Basic Properties

In this section, we formally provide our working assumptions. We assume access to multi-objective feedback with $m$ objectives $F(\mathbf{x}) = (f_1(\mathbf{x}), \ldots, f_m(\mathbf{x}))$, when $\forall i \in [m]$ the function $f_i(\mathbf{x})$ is smooth and self-concordant. Considering two scenarios AMOO, and $\epsilon$-AAMOO. In AMOO we assume the functions are aligned in the sense of Eq. (1), namely, that they share an optimal solution. In $\epsilon$-AAMOO we assume there is a (non-empty) set of $\epsilon$ approximate solutions in the sense of Eq. (7), i.e. every solution in the set has a maximum value gap of $\epsilon$ from the minima of every function.

We define the following quantities, for the single and multi objective settings:

$$\|\mathbf{y}\|_{\mathbf{x}}^2 := \|\mathbf{y}\|_{\nabla^2 f(\mathbf{x})} := \langle \nabla^2 f(\mathbf{x})\mathbf{y}, \mathbf{y} \rangle$$
$$\|\mathbf{y}\|_{\mathbf{x},\mathbf{w}}^2 := \|\mathbf{y}\|_{\nabla^2 f_{\mathbf{w}}(\mathbf{x})} := \langle \nabla^2 f_{\mathbf{w}}(\mathbf{x})\mathbf{y}, \mathbf{y} \rangle.$$

A smooth function satisfies the following:

**Lemma C.1** (Standard result, E.g., 9.17 Boyd and Vandenberghe (2004))**.** *Let $f : \mathbb{R}^n \to \mathbb{R}$ a $\beta$-smooth over $\mathbb{R}^n$, and let $\mathbf{x}_\star \in \arg\min_{\mathbf{x}} f(\mathbf{x})$. Then, for every $\mathbf{x} \in \mathbb{R}^n$ it holds that*

$$\|\nabla f(\mathbf{x})\|^2 \leq 2\beta \left( f(\mathbf{x}) - f(\mathbf{x}_\star) \right).$$

A self-concordant function satisfies the following:

**Lemma C.2** (Theorem 5.1.8 & Lemma 5.1.5, Nesterov (2013))**.** *Let $f : \mathbb{R}^n \to \mathbb{R}$ be an $M_{\mathrm{f}}$ self-concordant function. Let $\mathbf{x}, \mathbf{y} \in \mathbb{R}^n$ , we have*

$$f(\mathbf{y}) \geq f(\mathbf{x}) + \langle \nabla f(\mathbf{x}), \mathbf{y} - \mathbf{x} \rangle + \frac{1}{M_{\mathrm{f}}^2}\omega\left( M_{\mathrm{f}} \|\mathbf{y} - \mathbf{x}\|_{\mathbf{x}} \right),$$

*where $\omega(t) := t - \ln(1 - t)$, and, for any $t > 0$, $\omega(t) \geq \frac{t^2}{2(1+t)}$.*

**Lemma C.3** (Theorem 5.1.1, Nesterov et al. (2018))**.** *Let $f_1, f_2 : \mathbb{R}^n \to \mathbb{R}$ be $M_{\mathrm{f}}$ self-concordant functions. Let $w_1, w_2 > 0$. Then, $f = w_1 f_1 + w_2 f_2$ is $M = \max_i\{\frac{1}{\sqrt{w_i}}\}M_{\mathrm{f}}$ self-concordant function.*

**Proposition C.1** (Weighted sum of self-concordant functions)**.** *Let $\{f_i : \mathbb{R}^n \to \mathbb{R}\}_{i=1}^n$ be $M_{\mathrm{f}}$ self-concordant functions. Let $\{w_i > 0\}$. Then, $f = \sum_{i=1}^n w_i f_i$ is $M = \max_i\{\frac{1}{\sqrt{w_i}}\}M_{\mathrm{f}}$ self-concordant function.*

The proof of the last proposition proof is found in Section F.

**Theorem C.1** (Weyl's Theorem)**.** *Let $\mathbf{A}$ and $\Delta$ be symmetric matrices in $\mathbb{R}^{n \times n}$. Let $\lambda_j(\mathbf{A})$ be the $j^{th}$ largest eigenvalue of a matrix $\mathbf{A}$. Then, for all $j \in [n]$ it holds that $\|\lambda_j(\mathbf{A}) - \lambda_j(\mathbf{A} + \Delta)\| \leq \|\Delta\|_2$, where $\|\Delta\|_2$ is the spectral norm of $\Delta$.*

### C.1. Auxiliary Results

Further, we have the following simple consequence of the AMOO setting.

**Lemma C.4.** *For all $\mathbf{w} \in \Delta_m$ and $\mathbf{x} \in \mathbb{R}^n$ it holds that $f_{\mathbf{w}}(\mathbf{x}) - f_{\mathbf{w}}(\mathbf{x}_\star) \geq 0$.*

*Proof.* Observe that $f_{\mathbf{w}}(\mathbf{x}) - f_{\mathbf{w}}(\mathbf{x}_\star) = \sum_{i=1}^m w_i \left( f_i(\mathbf{x}) - f_i(\mathbf{x}_\star) \right)$. Since $\mathbf{x}_\star$ is the optimal solution for all objectives it holds that $f_i(\mathbf{x}) - f_i(\mathbf{x}_\star) \geq 0$. The lemma follows from the fact $w_i \geq 0$ for all $i \in [m]$. $\square$

**Lemma C.5** (Recurrence bound AMOO). *Let $\{r_k\}_{k \geq 0}$ be a sequence of non-negative real numbers that satisfy the recurrence relation*

$$r_{k+1}^2 \leq r_k^2 - \alpha_1 \frac{r_k^2}{1 + \alpha_2 r_k},$$

*where $\alpha_1 \in [0, 2)$ and $\alpha_2 \in \mathbb{R}_+$. Let $k_0 := \left\lceil \frac{4(r_0 \alpha_2 - 1)}{\alpha_1} \right\rceil$. Then, $r_k$ is bounded by*

$$r_k \leq \begin{cases} r_{k_0} \left(1 - \frac{\alpha_1}{2}\right)^{\frac{k - k_0}{2}} & k \geq k_0 \\ r_0 - \frac{\alpha_1}{4\alpha_2} k & o.w. \end{cases}.$$

The proof of the lemma is found in Appendix F.3

The next lemma is for the $\epsilon$-AMOO setting. We separate them since the recursion and the result of $\epsilon$-AMOO have more details, making it less readable.

**Lemma C.6** (Recurrence bound $\epsilon$-AMOO). *Let $\{r_k\}_{k \geq 0}$ be a sequence of non-negative real numbers that satisfy the recurrence relation*

$$r_{k+1}^2 \leq r_k^2 - \alpha_1 \frac{r_k^2}{1 + \alpha_2 r_k} + \alpha_3 + \alpha_4 r_k,$$

*where $\alpha_1 \in (0, 2)$, $\alpha_2 \in \mathbb{R}_+$, $\alpha_3 \leq \frac{\alpha_1^2}{256\alpha_2^2}$, and $\alpha_4 \leq \frac{\alpha_1}{4\alpha_2}$. Let $k_0 := \left\lceil \frac{16(r_0 \alpha_2 - 1)}{\alpha_1} \right\rceil$. Then, $r_k$ is bounded by*

$$r_k \leq \begin{cases} r_{k_0} \left(1 - \frac{\alpha_1}{2}\right)^{\frac{k' - k}{2}} + \sqrt{\frac{2\alpha_3}{\alpha_1} + \frac{2\alpha_4}{\alpha_1 \alpha_2}} & k \geq k_0 \\ r_0 - \frac{\alpha_1}{16\alpha_2} k & o.w. \end{cases}.$$

# D. Proofs of AMOO Results

See Section 4 for a highlevel description of key steps in the proof.

We first describe two useful lemmas that are used across the section.

**Lemma D.1** (Weighted Function is Self-Concordant). *For any iteration $k$ of `CAMOO`, the function $f_{\mathbf{w}_k}$ is $1/\sqrt{w_{\min}} M_{\mathrm{f}}$ self-concordant.*

*Proof.* This is a direct consequence of Proposition C.1 and the fact Algorithm 1 sets the weights by optimizing over a set where the weight vector is lower bounded by $w_{\min}$. □

**Lemma D.2** (Continuity of Minimal Eigenvalue of Hessian). *Let $\mathbf{x} \in \mathbb{R}^n$. Further, let $\mathbf{w}_\star \in \arg\max_{\mathbf{w} \in \Delta_m} \lambda_{\min} \left(\nabla^2 f_{\mathbf{w}}(\mathbf{x})\right)$, $\widehat{\mathbf{w}} \in \arg\max_{\mathbf{w} \in \Delta_{m, w_{\min}}} \lambda_{\min} \left(\nabla^2 f_{\mathbf{w}}(\mathbf{x})\right)$. It holds that $\lambda_{\min} \left(\nabla^2 f_{\widehat{\mathbf{w}}}(\mathbf{x})\right) \geq \lambda_{\min} \left(\nabla^2 f_{\mathbf{w}_\star}(\mathbf{x})\right) - 2m w_{\min} \beta$.*

With these two results, we are ready to prove Theorem 4.2 and Theorem 5.2.

## D.1. Proof of Theorem 4.2

Restate it first:

**Theorem 4.2** ($\mu_{\mathrm{G}}$ Convergence of `CAMOO`). *Suppose $\{f_i\}_{i \in [m]}$ are $\beta$ smooth, $M_{\mathrm{f}}$ self-concordant, share an optimal solution $\mathbf{x}_\star$ and that $\mu_{\mathrm{G}} > 0$. Let $k_0 := \left\lceil \frac{16\beta \left(\|\mathbf{x}_0 - \mathbf{x}_\star\| 3\sqrt{m} \beta M_{\mathrm{f}} - \sqrt{\mu_{\mathrm{G}}}\right)}{3\mu_{\mathrm{G}}^{3/2}} \right\rceil$, where $\|\cdot\|$ is the Euclidean-norm. Then, `Weighted-GD` instantiated with `CAMOO` weight-optimizer and $\eta = 1/2\beta$ converges with rate:*

$$\|\mathbf{x}_k - \mathbf{x}_\star\| \leq \begin{cases} \|\mathbf{x}_{k_0} - \mathbf{x}_\star\| \left(1 - \frac{3\mu_{\mathrm{G}}}{8\beta}\right)^{(k - k_0)/2} & k \geq k_0 \\ \|\mathbf{x}_0 - \mathbf{x}_\star\| - k \frac{\mu_{\mathrm{G}}^{3/2}}{16\beta^2 \sqrt{m} M_{\mathrm{f}}} & o.w. \end{cases}$$

*Proof.* At each iteration Algorithm 1 gets $\mathbf{w}_k \in \underset{\mathbf{w} \in \Delta_{m,w_{\min}}}{\arg\max} \lambda_{\min} \left( \sum_i w_i \nabla^2 f_i(\mathbf{x}_k) \right)$. Using the assumption that $\underset{\mathbf{w} \in \Delta_m}{\max} \lambda_{\min} \left( \nabla^2 f_{\mathbf{w}_k}(\mathbf{x}_k) \right) \geq \mu_{\mathrm{G}}$, Lemma D.2, and since we set $w_{\min} = \mu_{\mathrm{G}} / (8m\beta)$ we have that

$$\lambda_{\min} \left( \nabla^2 f_{\mathbf{w}_k} \right) \geq \lambda_{\min} \left( \nabla^2 f_{\mathbf{w}} \right) - 2m w_{\min}\beta \geq \mu_{\mathrm{G}} - \mu_{\mathrm{G}}/4 = (3/4)\mu_{\mathrm{G}}, \tag{12}$$

for all iterations $t$. Recall that the update rule is given by $\mathbf{x}_{k+1} = \mathbf{x}_k - \eta \nabla f_{\mathbf{w}_k}(\mathbf{x}_k)$, where $\eta$ is the step size. Then, for every $\mathbf{x} \in \mathbb{R}^n$ we have

$$\begin{aligned}
\|\mathbf{x}_{k+1} - \mathbf{x}\|^2 &= \|\mathbf{x}_k - \eta \nabla f_{\mathbf{w}_k}(\mathbf{x}_k) - \mathbf{x}\|^2 \\
&= \|\mathbf{x}_k - \mathbf{x}\|^2 - 2\eta \langle \nabla f_{\mathbf{w}_k}(\mathbf{x}_k), \mathbf{x}_k - \mathbf{x} \rangle + \eta^2 \|\nabla f_{\mathbf{w}_k}(\mathbf{x}_k)\|^2.
\end{aligned} \tag{13}$$

By Lemma D.1 it holds that $f_{\mathbf{w}_k}$ is

$$\widehat{M_{\mathrm{f}}} := 1/\sqrt{w_{\min}} M_{\mathrm{f}} \leq 3\sqrt{m\beta} M_{\mathrm{f}}/\sqrt{\mu_{\mathrm{G}}} \tag{14}$$

self-concordant. Then, from Lemma C.2, by properties of self-concordant functions, for every $\mathbf{x} \in \mathbb{R}^n$ we have

$$\langle \nabla f_{\mathbf{w}_k}(\mathbf{x}_k), \mathbf{x}_k - \mathbf{x} \rangle \geq f_{\mathbf{w}_k}(\mathbf{x}_k) - f_{\mathbf{w}_k}(\mathbf{x}) + \frac{1}{\widehat{M_{\mathrm{f}}}^2} \omega \left( \widehat{M_{\mathrm{f}}} \|\mathbf{x} - \mathbf{x}_k\|_{\mathbf{x}_k, \mathbf{w}_k} \right),$$

Plugging this inequality into Eq. (13) implies that

$$\|\mathbf{x}_{k+1} - \mathbf{x}\|^2 \leq \|\mathbf{x}_k - \mathbf{x}\|^2 - 2\eta \left( f_{\mathbf{w}_k}(\mathbf{x}_k) - f_{\mathbf{w}_k}(\mathbf{x}) + \frac{1}{\widehat{M_{\mathrm{f}}}^2} \omega \left( \widehat{M_{\mathrm{f}}} \|\mathbf{x} - \mathbf{x}_k\|_{\mathbf{x}_k, \mathbf{w}_k} \right) \right) + \eta^2 \|\nabla f_{\mathbf{w}_k}(\mathbf{x}_k)\|^2. \tag{15}$$

Recall that $\mathbf{x}_\star \in \arg\min_{\mathbf{x}} f_i(\mathbf{x})$ for every $i \in [m]$. Since $f_{\mathbf{w}_k}$ is $\beta$-smooth, by using Lemma C.1, and plugging in $\mathbf{x}_\star$ we have

$$\begin{aligned}
&\|\mathbf{x}_{k+1} - \mathbf{x}_\star\|^2 \\
&\leq \|\mathbf{x}_k - \mathbf{x}_\star\|^2 - 2\eta \left( f_{\mathbf{w}_k}(\mathbf{x}_k) - f_{\mathbf{w}_k}(\mathbf{x}_\star) + \frac{1}{\widehat{M_{\mathrm{f}}}^2} \omega \left( \widehat{M_{\mathrm{f}}} \|\mathbf{x}_\star - \mathbf{x}_k\|_{\mathbf{x}_k, \mathbf{w}_k} \right) \right) + 2\beta\eta^2 \left( f_{\mathbf{w}_k}(\mathbf{x}_k) - f_{\mathbf{w}_k}(\mathbf{x}_\star) \right) \\
&= \|\mathbf{x}_k - \mathbf{x}_\star\|^2 - 2\eta \frac{1}{\widehat{M_{\mathrm{f}}}^2} \omega \left( \widehat{M_{\mathrm{f}}} \|\mathbf{x}_\star - \mathbf{x}_k\|_{\mathbf{x}_k, \mathbf{w}_k} \right) + 2\eta(\beta\eta - 1) \left( f_{\mathbf{w}_k}(\mathbf{x}_k) - f_{\mathbf{w}_k}(\mathbf{x}_\star) \right).
\end{aligned}$$

By using Lemma C.4 it holds that $f_{\mathbf{w}_k}(\mathbf{x}_k) - f_{\mathbf{w}_k}(\mathbf{x}_\star) \geq 0$, and since $0 < \eta \leq 1/2\beta$, it holds that $2\eta(\beta\eta - 1) \left( f_{\mathbf{w}_k}(\mathbf{x}_k) - f_{\mathbf{w}_k}(\mathbf{x}_\star) \right) \leq 0$. Then, by using the lower bound from Lemma C.2, i.e. $\omega(t) \geq \frac{t^2}{2(1+t)}$, the following holds

$$\|\mathbf{x}_{k+1} - \mathbf{x}_\star\|^2 \leq \|\mathbf{x}_k - \mathbf{x}_\star\|^2 - 2\eta \frac{\|\mathbf{x}_\star - \mathbf{x}_k\|_{\mathbf{x}_k, \mathbf{w}_k}^2}{1 + \widehat{M_{\mathrm{f}}} \|\mathbf{x}_\star - \mathbf{x}_k\|_{\mathbf{x}, \mathbf{w}_k}}.$$

Note that $\lambda_{\min} \left( \nabla^2 f_{\mathbf{w}_k}(\mathbf{x}_k) \right) \|\mathbf{x}_\star - \mathbf{x}_k\|^2 \leq \|\mathbf{x}_\star - \mathbf{x}_k\|_{\mathbf{x}_k, \mathbf{w}_k}^2 \leq \lambda_{\max} \left( \nabla^2 f_{\mathbf{w}_k}(\mathbf{x}_k) \right) \|\mathbf{x}_\star - \mathbf{x}_k\|^2$. By using the following: $\lambda_{\min} \left( \nabla^2 f_{\mathbf{w}_k}(\mathbf{x}_k) \right) \geq (3/4)\mu_{\mathrm{G}}$ (Eq. (12)), $\lambda_{\max} \left( \nabla^2 f_{\mathbf{w}_k}(\mathbf{x}_k) \right) \leq \beta$ (smoothness), $\widehat{M_{\mathrm{f}}} \leq 3\sqrt{m\beta} M_{\mathrm{f}}/\sqrt{\mu_{\mathrm{G}}}$ (Eq. (14)), and $\eta = 1/2\beta$, we obtain

$$\|\mathbf{x}_{k+1} - \mathbf{x}_\star\|^2 \leq \|\mathbf{x}_k - \mathbf{x}_\star\|^2 - \frac{3\mu_{\mathrm{G}}}{4\beta} \frac{\|\mathbf{x}_\star - \mathbf{x}_k\|^2}{1 + \left( 3\sqrt{m\beta} M_{\mathrm{f}}/\sqrt{\mu_{\mathrm{G}}} \right) \|\mathbf{x}_\star - \mathbf{x}_k\|}.$$

Now, we are ready for the last step. Denote $\alpha_1 = \frac{3\mu_{\mathrm{G}}}{4\beta}$, and $\alpha_2 = \frac{3\sqrt{m\beta} M_{\mathrm{f}}}{\sqrt{\mu_{\mathrm{G}}}}$. Since $\alpha_1 \in (0, 1]$, $\alpha_2 \in \mathbb{R}_+$, and $\|\mathbf{x} - \mathbf{x}_\star\| \in \mathbb{R}_+$ for every $\mathbf{x} \in \mathbb{R}^n$, we arrive to the recurrence relation analyzed in Lemma C.5. Then, we obtain

$$\|\mathbf{x}_k - \mathbf{x}_\star\| \leq \begin{cases} \|\mathbf{x}_{k_0} - \mathbf{x}_\star\| \left( 1 - \frac{3\mu_{\mathrm{G}}}{8\beta} \right)^{(k-k_0)/2} & k \geq k_0 \\ \|\mathbf{x}_0 - \mathbf{x}_\star\| - k \frac{\mu_{\mathrm{G}}^{3/2}}{16\beta^2 \sqrt{m} M_{\mathrm{f}}} & o.w. \end{cases}$$

where $k_0 := \left\lceil \frac{16\beta \left( \|\mathbf{x}_0 - \mathbf{x}_\star\| 3\sqrt{m}\beta M_{\mathrm{f}} - \sqrt{\mu_{\mathrm{G}}} \right)}{3\mu_{\mathrm{G}}^{3/2}} \right\rceil$. $\qquad\square$

## D.2. Proof of Theorem 5.2

Restate it first:

**Theorem 5.2** ($\mu_L$ Convergence of PAMOO). *Suppose $\{f_i\}_{i\in[m]}$ are $\beta$ smooth, $M_f$ self-concordant, share an optimal solution $\mathbf{x}_\star$ and $\mu_L > 0$. Let $k_0 := \left\lceil \frac{64\beta\left(\|\mathbf{x}_0 - \mathbf{x}_\star\|3\sqrt{m}\beta M_f - \sqrt{\mu_L}\right)}{3\mu_L^{3/2}} \right\rceil$, where $\|\cdot\|$ is the Euclidean-norm. Then, Weighted-GD instantiated with PAMOO weight-optimizer and $\eta = 1$ converges with rate:*

$$\|\mathbf{x}_k - \mathbf{x}_\star\| \leq \begin{cases} \|\mathbf{x}_{k_0} - \mathbf{x}_\star\| \left(1 - \frac{3\mu_L}{32\beta}\right)^{(k-k_0)/2} & k \geq k_0 \\ \|\mathbf{x}_0 - \mathbf{x}_\star\| - k\frac{\mu_L^{3/2}}{64\beta^2\sqrt{m}M_f} & o.w. \end{cases}$$

*Proof.* Recall that for every $\mathbf{w} \in \mathbb{R}_+^m$ it holds that $f_\mathbf{w}$ is a convex function. Hence, for every $\mathbf{x}, \mathbf{y} \in \mathbb{R}^n$ it holds that

$$f_\mathbf{w}(\mathbf{x}) - f_\mathbf{w}(\mathbf{y}) \leq \nabla f_\mathbf{w}(\mathbf{x})^T(\mathbf{x} - \mathbf{y}).$$

Recall that the step size $\eta = 1$, then the update rule is given by $\mathbf{x}_{k+1} = \mathbf{x}_k - \nabla f_{\mathbf{w}_k}(\mathbf{x}_k)$. Then, by using the previous equation, for every $\mathbf{x} \in \mathbb{R}^n$ we have

$$\begin{aligned} \|\mathbf{x}_{k+1} - \mathbf{x}\|^2 &= \|\mathbf{x}_k - \mathbf{x}\|^2 - 2\langle\nabla f_{\mathbf{w}_k}(\mathbf{x}_k), \mathbf{x}_k - \mathbf{x}\rangle + \|\nabla f_{\mathbf{w}_k}(\mathbf{x}_k)\|^2 \\ &\leq \|\mathbf{x}_k - \mathbf{x}\|^2 - 2\left(f_{\mathbf{w}_k}(\mathbf{x}_k) - f_{\mathbf{w}_k}(\mathbf{x})\right) + \|\nabla f_{\mathbf{w}_k}(\mathbf{x}_k)\|^2, \end{aligned} \tag{16}$$

Recall that $\mathbf{x}_\star \in \arg\min_{\mathbf{x}\in\mathbb{R}^n} f_i(\mathbf{x})$ for every $i \in [m]$. Since the update rule of PAMOO is

$$\mathbf{w}_k \in \arg\max_{\mathbf{w}\in\mathbb{R}_+^m} 2\left(f_{\mathbf{w}_k}(\mathbf{x}_k) - f_{\mathbf{w}_k}(\mathbf{x}_\star)\right) - \|\nabla f_{\mathbf{w}_k}(\mathbf{x}_k)\|^2,$$

the following holds

$$\|\mathbf{x}_{k+1} - \mathbf{x}_\star\|^2 \leq \|\mathbf{x}_k - \mathbf{x}_\star\|^2 - \max_{\mathbf{w}\in\mathbb{R}_+^m}\left\{2\left(f_\mathbf{w}(\mathbf{x}_k) - f_\mathbf{w}(\mathbf{x}_\star)\right) - \|\nabla f_\mathbf{w}(\mathbf{x}_k)\|^2\right\}, \tag{17}$$

Denote $\mathbf{w}_\star = \arg\max_{\mathbf{w}\in\Delta_{m,w_{\min}}} \lambda_{\min}\left(\sum_{i=1}^m w_i\nabla^2 f_i(\mathbf{x}_\star)\right)$, $a_k = f_{\mathbf{w}_\star}(\mathbf{x}_k) - f_{\mathbf{w}_\star}(\mathbf{x}_\star)$, and $b_k = \|\nabla f_{\mathbf{w}_\star}(\mathbf{x}_k)\|^2$. Let $w(\mathbf{x}_k) = \mathbf{w}_\star\frac{a_k}{b_k} \in \mathbb{R}_+^m$, we can lower bound of the last expression as follows

$$\begin{aligned} \max_{\mathbf{w}\in\mathbb{R}_+^m}\left[2\left(f_\mathbf{w}(\mathbf{x}_k) - f_\mathbf{w}(\mathbf{x}_\star)\right) - \|\nabla f_\mathbf{w}(\mathbf{x}_k)\|^2\right] &\geq 2\left(f_{\mathbf{w}(\mathbf{x}_k)}(\mathbf{x}_k) - f_{\mathbf{w}(\mathbf{x}_k)}(\mathbf{x}_\star)\right) - \left\|\nabla f_{\mathbf{w}(\mathbf{x}_k)}(\mathbf{x}_k)\right\|^2 \\ &= \frac{\left(f_{\mathbf{w}_\star}(\mathbf{x}_k) - f_{\mathbf{w}_\star}(\mathbf{x}_\star)\right)^2}{\|\nabla f_{\mathbf{w}_\star}(\mathbf{x}_k)\|^2}. \end{aligned}$$

Since $\mathbf{w}_\star \in \Delta_{m,w_{\min}}$ it holds that $f_{\mathbf{w}_\star}$ is $\beta$ smooth. Then, it holds that $\|\nabla f_{\mathbf{w}_\star}(\mathbf{x})\|^2 \leq 2\beta\left(f_{\mathbf{w}_\star}(\mathbf{x}) - f_{\mathbf{w}_\star}(\mathbf{x}_\star)\right)$ for every $\mathbf{x}$, and we have

$$\max_{\mathbf{w}\in\mathbb{R}_+^m}\left[2\left(f_\mathbf{w}(\mathbf{x}_k) - f_\mathbf{w}(\mathbf{x}_\star)\right) - \|\nabla f_\mathbf{w}(\mathbf{x}_k)\|^2\right] \geq \frac{f_{\mathbf{w}_\star}(\mathbf{x}_k) - f_{\mathbf{w}_\star}(\mathbf{x}_\star)}{2\beta}$$

Plugging this in Eq. (17), we arrive to

$$\|\mathbf{x}_{k+1} - \mathbf{x}_\star\|^2 \leq \|\mathbf{x}_k - \mathbf{x}_\star\|^2 - \frac{f_{\mathbf{w}_\star}(\mathbf{x}_k) - f_{\mathbf{w}_\star}(\mathbf{x}_\star)}{2\beta}$$

By Lemma D.1 it holds that $f_{\mathbf{w}_\star}$ is $\widehat{M_f} := 1/\sqrt{w_{\min}}M_f$ self concordant. Then, From Lemma C.2, and its lower bound , i.e. $\omega(t) \geq \frac{t^2}{2(1+t)}$, we have

$$f_{\mathbf{w}_\star}(\mathbf{x}_k) - f_{\mathbf{w}_\star}(\mathbf{x}_\star) \geq \langle\nabla f_{\mathbf{w}_\star}(\mathbf{x}_\star), \mathbf{x}_k - \mathbf{x}_\star\rangle + \frac{\|\mathbf{x}_k - \mathbf{x}_\star\|_{\mathbf{x}_\star,\mathbf{w}_\star}^2}{2\left(1 + \widehat{M_f}\|\mathbf{x}_k - \mathbf{x}_\star\|_{\mathbf{x}_\star,\mathbf{w}_\star}\right)} = \frac{\|\mathbf{x}_k - \mathbf{x}_\star\|_{\mathbf{x}_\star,\mathbf{w}_\star}^2}{2\left(1 + \widehat{M_f}\|\mathbf{x}_k - \mathbf{x}_\star\|_{\mathbf{x}_\star,\mathbf{w}_\star}\right)}.$$

The equality is due to the optimality condition, $\nabla f_i(\mathbf{x}_\star) = 0$ for every $i \in [m]$, thus, $\nabla f_{\mathbf{w}_\star}(\mathbf{x}_\star) = \mathbf{0}$. Combining the last two equations, we have

$$\|\mathbf{x}_{k+1} - \mathbf{x}_\star\|^2 \le \|\mathbf{x}_k - \mathbf{x}_\star\|^2 - \frac{1}{4\beta} \frac{\|\mathbf{x}_k - \mathbf{x}_\star\|_{\mathbf{x}_\star, \mathbf{w}_\star}^2}{1 + \widehat{M}_{\mathrm{f}} \|\mathbf{x}_k - \mathbf{x}_\star\|_{\mathbf{x}_\star, \mathbf{w}_\star}}$$

Note that $\lambda_{\min}\left(\nabla^2 f_{\mathbf{w}_\star}(\mathbf{x}_\star)\right) \|\mathbf{x}_\star - \mathbf{x}_k\|^2 \le \|\mathbf{x}_\star - \mathbf{x}_k\|_{\mathbf{x}_\star, \mathbf{w}_\star}^2 \le \lambda_{\max}\left(\nabla^2 f_{\mathbf{w}_\star}(\mathbf{x}_\star)\right) \|\mathbf{x}_\star - \mathbf{x}_k\|^2$. Since $\lambda_{\max}\left(\nabla^2 f_{\mathbf{w}_\star}(\mathbf{x}_\star)\right) \le \beta$ (smoothness), and since $w_{\min} = \mu_{\mathrm{L}}/(8m\beta)$, then $\widehat{M}_{\mathrm{f}} \le 3\sqrt{m}\beta M_{\mathrm{f}}/\sqrt{\mu_{\mathrm{L}}}$. Thus, it holds that

$$\|\mathbf{x}_{k+1} - \mathbf{x}_\star\|^2 \le \|\mathbf{x}_k - \mathbf{x}_\star\|^2 - \frac{1}{4\beta} \frac{\lambda_{\min}\left(\nabla^2 f_{\mathbf{w}_\star}(\mathbf{x}_\star)\right) \|\mathbf{x}_\star - \mathbf{x}_k\|^2}{1 + \frac{3\sqrt{m}M_{\mathrm{f}}\beta}{\sqrt{\mu_{\mathrm{L}}}} \|\mathbf{x}_k - \mathbf{x}_\star\|}$$

Using the assumption that $\max_{\mathbf{w} \in \Delta_m} \lambda_{\min}\left(\nabla^2 f_{\mathbf{w}}(\mathbf{x}_\star)\right) \ge \mu_{\mathrm{L}}$, Lemma D.2, we have that

$$\lambda_{\min}\left(\nabla^2 f_{\mathbf{w}_\star}(\mathbf{x}_\star)\right) \ge \max_{\mathbf{w} \in \Delta_m} \lambda_{\min}\left(\nabla^2 f_{\mathbf{w}}(\mathbf{x}_\star)\right) - 2m w_{\min}\beta \ge \mu_{\mathrm{L}} - \mu_{\mathrm{L}}/4 = (3/4)\mu_{\mathrm{L}}.$$

Finally, we obtain the recurring equation we wish:

$$\|\mathbf{x}_{k+1} - \mathbf{x}_\star\|^2 \le \|\mathbf{x}_k - \mathbf{x}_\star\|^2 - \frac{3\mu_{\mathrm{L}}}{16\beta} \frac{\|\mathbf{x}_\star - \mathbf{x}_k\|^2}{1 + \frac{3\sqrt{m}M_{\mathrm{f}}\beta}{\sqrt{\mu_{\mathrm{L}}}} \|\mathbf{x}_k - \mathbf{x}_\star\|}$$

Denote $\alpha_1 = \frac{3\mu_{\mathrm{L}}}{16\beta}$, and $\alpha_2 = \frac{3\sqrt{m}\beta M_{\mathrm{f}}}{\sqrt{\mu_{\mathrm{L}}}}$. Note that $\alpha_1 \in (0, 1]$, $\alpha_2 \in \mathbb{R}_+$, and $\|\mathbf{x} - \mathbf{x}_\star\| \in \mathbb{R}_+$ for every $\mathbf{x} \in \mathbb{R}^n$, we arrive to the recurrence relation analyzed in Lemma C.5. Then, we obtain

$$\|\mathbf{x}_k - \mathbf{x}_\star\| \le \begin{cases} \|\mathbf{x}_{k_0} - \mathbf{x}_\star\| \left(1 - \frac{3\mu_{\mathrm{L}}}{32\beta}\right)^{(k-k_0)/2} & k \ge k_0 \\ \|\mathbf{x}_0 - \mathbf{x}_\star\| - k\frac{\mu_{\mathrm{L}}^{3/2}}{64\beta^2\sqrt{m}M_{\mathrm{f}}} & o.w. \end{cases}$$

where $k_0 := \left\lceil \frac{64\beta\left(\|\mathbf{x}_0 - \mathbf{x}_\star\|3\sqrt{m}\beta M_{\mathrm{f}} - \sqrt{\mu_{\mathrm{L}}}\right)}{3\mu_{\mathrm{L}}^{3/2}} \right\rceil$. $\qquad \square$

# E. $\epsilon$-Approximation Solution

In this section, we represent the formal theorems of the Informal Theorem 6.2 for `CAMOO` and `PAMOO`. First, we rewrite the definition of $\epsilon$- approximate solutions set.

**Definition E.1** ($\epsilon$-Approximate Solution Set). Let $\epsilon \ge 0$. $\mathcal{C}_\epsilon$ is $\epsilon$-Approximate Solution Set ($\epsilon$-ASS) if for every $i \in [m]$ it holds that $f_i(\mathbf{x}) - f_i(\mathbf{x}_\star^i) \le \epsilon$, i.e.

$$\mathcal{C}_\epsilon = \{\mathbf{x} \in \mathbb{R}^n | f_i(\mathbf{x}) - f_i(\mathbf{x}_\star^i) \le \epsilon \ \forall i \in [m]\},$$

where $\mathbf{x}_\star^i \in \arg\min_{\mathbf{x} \in \mathbb{R}^n}\{f_i(\mathbf{x})\}$.

We show in the following formal theorem that for $\epsilon > 0$ `CAMOO` converges for any chosen point from the set $\mathcal{C}_\epsilon$, i.e. the $\epsilon$-approximate solutions set.

**Theorem E.2** ($\mu_{\mathrm{G}}$ Approximation Convergence of `CAMOO`). *Suppose $\{f_i\}_{i \in [m]}$ are $\beta$ smooth, $M_{\mathrm{f}}$ self-concordant, $\mathcal{C}_\epsilon$ is an $\epsilon$-ASS and that $\mu_{\mathrm{G}} > 0$. Let $k_0 := \left\lceil \frac{64\beta\left(\frac{3\sqrt{m}\beta M_{\mathrm{f}}\|\mathbf{x}_0 - \mathbf{x}_\star^\epsilon\|}{\sqrt{\mu_{\mathrm{G}}}} - 1\right)}{3\mu_{\mathrm{G}}} \right\rceil$, where $\|\cdot\|$ is the 2-norm. Let $\frac{\mu_{\mathrm{G}}^3}{4^6 m\beta^3 M_{\mathrm{f}}^2} \ge \epsilon > 0$. Then, for every point $\mathbf{x}_\star^\epsilon \in \mathcal{C}_\epsilon$, `Weighted-GD` instantiated with `CAMOO` weight-optimizer and $\eta = 1/2\beta$ converges with rate:*

$$\|\mathbf{x}_k - \mathbf{x}_\star^\epsilon\| \le \begin{cases} \|\mathbf{x}_{k_0} - \mathbf{x}_\star^\epsilon\| \left(1 - \frac{3\mu_{\mathrm{G}}}{8\beta}\right)^{(k-k_0)/2} + \sqrt{\frac{8}{3\mu_{\mathrm{G}}}\epsilon} & k \ge k_0 \\ \|\mathbf{x}_0 - \mathbf{x}_\star^\epsilon\| - k\frac{\mu_{\mathrm{G}}^{3/2}}{4^3\beta^2\sqrt{m}M_{\mathrm{f}}} & o.w. \end{cases}$$

*Proof.* Let $\mathbf{x}_\star^\epsilon \in \mathcal{C}_\epsilon$. Denote $\mathbf{x}_k^\star = \arg\min_{\mathbf{x}\in\mathbb{R}^n} f_{\mathbf{w}_k}(\mathbf{x})$ for every $k$, then since $f_{\mathbf{w}_k}$ is $\beta$-smooth, by using Lemma C.1, it holds that $\|\nabla f(\mathbf{x})\|^2 \le 2\beta\left(f(\mathbf{x}) - f(\mathbf{x}_\star)\right)$. Plugging in $\mathbf{x} = \mathbf{x}_\star^\epsilon$ in Eq. (15), we have

$$\|\mathbf{x}_{k+1} - \mathbf{x}_\star^\epsilon\|^2$$

$$\le \|\mathbf{x}_k - \mathbf{x}_\star^\epsilon\|^2 - 2\eta\left(f_{\mathbf{w}_k}(\mathbf{x}_k) - f_{\mathbf{w}_k}(\mathbf{x}_\star^\epsilon) + \frac{1}{\widehat{M_\mathrm{f}}^2}\omega\left(\widehat{M_\mathrm{f}}\|\mathbf{x}_\star^\epsilon - \mathbf{x}_k\|_{\mathbf{x}_k,\mathbf{w}_k}\right)\right) + 2\beta\eta^2\left(f_{\mathbf{w}_k}(\mathbf{x}_k) - f_{\mathbf{w}_k}(\mathbf{x}_k^\star)\right)$$

$$= \|\mathbf{x}_k - \mathbf{x}_\star^\epsilon\|^2 - 2\eta\frac{1}{\widehat{M_\mathrm{f}}^2}\omega\left(\widehat{M_\mathrm{f}}\|\mathbf{x}_\star^\epsilon - \mathbf{x}_k\|_{\mathbf{x}_k,\mathbf{w}_k}\right)$$

$$+ 2\eta\left(f_{\mathbf{w}_k}(\mathbf{x}_\star^\epsilon) - f_{\mathbf{w}_k}(\mathbf{x}_k^\star)\right) + 2\eta(\beta\eta - 1)\left(f_{\mathbf{w}_k}(\mathbf{x}_k) - f_{\mathbf{w}_k}(\mathbf{x}_k^\star)\right).$$

Since $f_{\mathbf{w}_k}(\mathbf{x}_k) - f_{\mathbf{w}_k}(\mathbf{x}_k^\star) \ge 0$, and since $0 < \eta \le 1/2\beta$, it holds that $2\eta(\beta\eta - 1)\left(f_{\mathbf{w}_k}(\mathbf{x}_k) - f_{\mathbf{w}_k}(\mathbf{x}_k^\star)\right) \le 0$. In addition, by using the lower bound from Lemma C.2, i.e. $\omega(t) \ge \frac{t^2}{2(1+t)}$, and Definition E.1 with the fact that $\sum_i w_i = 1$, the following holds

$$\|\mathbf{x}_{k+1} - \mathbf{x}_\star^\epsilon\|^2 \le \|\mathbf{x}_k - \mathbf{x}_\star^\epsilon\|^2 - 2\eta\frac{\|\mathbf{x}_\star^\epsilon - \mathbf{x}_k\|_{\mathbf{x}_k,\mathbf{w}_k}^2}{1 + \widehat{M_\mathrm{f}}\|\mathbf{x}_\star^\epsilon - \mathbf{x}_k\|_{\mathbf{x},\mathbf{w}_k}} + 2\eta\epsilon.$$

Note that $\lambda_{\min}\left(\nabla^2 f_{\mathbf{w}_k}(\mathbf{x}_k)\right)\|\mathbf{x}_\star^\epsilon - \mathbf{x}_k\|^2 \le \|\mathbf{x}_\star^\epsilon - \mathbf{x}_k\|_{\mathbf{x}_k,\mathbf{w}_k}^2 \le \lambda_{\max}\left(\nabla^2 f_{\mathbf{w}_k}(\mathbf{x}_k)\right)\|\mathbf{x}_\star^\epsilon - \mathbf{x}_k\|^2$. By using the following: $\lambda_{\min}\left(\nabla^2 f_{\mathbf{w}_k}(\mathbf{x}_k)\right) \ge (3/4)\mu_\mathrm{G}$ (Eq. (12)), $\lambda_{\max}\left(\nabla^2 f_{\mathbf{w}_k}(\mathbf{x}_k)\right) \le \beta$ (smoothness), $\widehat{M_\mathrm{f}} \le 3\sqrt{m\beta}M_\mathrm{f}/\sqrt{\mu_\mathrm{G}}$ (Eq. (14)), and $\eta = 1/2\beta$, we obtain

$$\|\mathbf{x}_{k+1} - \mathbf{x}_\star^\epsilon\|^2 \le \|\mathbf{x}_k - \mathbf{x}_\star^\epsilon\|^2 - \frac{3\mu_\mathrm{G}}{4\beta}\frac{\|\mathbf{x}_\star^\epsilon - \mathbf{x}_k\|^2}{1 + \left(3\sqrt{m\beta}M_\mathrm{f}/\sqrt{\mu_\mathrm{G}}\right)\|\mathbf{x}_\star^\epsilon - \mathbf{x}_k\|} + \frac{\epsilon}{\beta}.$$

Now, we are ready for the last step. Denote $\alpha_1 = \frac{3\mu_\mathrm{G}}{4\beta}$, $\alpha_2 = \frac{3\sqrt{m\beta}M_\mathrm{f}}{\sqrt{\mu_\mathrm{G}}}$, and $\alpha_3 = \frac{\epsilon}{\beta}$. Since $\alpha_1 \in (0,2)$, $\alpha_2 \in \mathbb{R}_+$, $\alpha_3 \le \frac{\alpha_1^2}{256\alpha_2^2}$ and $\|\mathbf{x} - \mathbf{x}_\star^\epsilon\| \in \mathbb{R}_+$ for every $\mathbf{x} \in \mathbb{R}^n$, we arrive to the recurrence relation analyzed in Lemma C.6. Then, we obtain

$$\|\mathbf{x}_k - \mathbf{x}_\star^\epsilon\| \le \begin{cases} \|\mathbf{x}_{k_0} - \mathbf{x}_\star^\epsilon\|\left(1 - \frac{3\mu_\mathrm{G}}{8\beta}\right)^{(k-k_0)/2} + \sqrt{\frac{8}{3\mu_\mathrm{G}}\epsilon} & k \ge k_0 \\ \|\mathbf{x}_0 - \mathbf{x}_\star^\epsilon\| - k\frac{\mu_\mathrm{G}^{3/2}}{4^3\beta^2\sqrt{m}M_\mathrm{f}} & o.w. \end{cases}$$

where $k_0 := \left\lceil \frac{64\beta\left(\frac{3\sqrt{m\beta}M_\mathrm{f}\|\mathbf{x}_0 - \mathbf{x}_\star^\epsilon\|}{\sqrt{\mu_\mathrm{G}}} - 1\right)}{3\mu_\mathrm{G}} \right\rceil$. $\qquad\square$

Before we continue to PAMOO, we recall Definition 6.1 which defines $\mu_\mathrm{L}^\epsilon$. This parameter is the maximum curvature over the $\epsilon$-approximate solutions set, which can be much greater than $\mu_\mathrm{G}$, and $\mu_\mathrm{L}$. Before we show the formal theorem of PAMOO, we define $\widehat{\mathbf{x}_\star^\epsilon} \in \mathcal{C}_\epsilon$ which is the point with the maximum curvature, s.t.

$$\widehat{\mathbf{x}_\star^\epsilon} = \arg\max_{\mathbf{x}_\star^\epsilon \in \mathcal{C}_\epsilon} \max_{\mathbf{w}\in\Delta_m} \lambda_{\min}\left(\sum_{i=1}^m w_i \nabla^2 f_i(\mathbf{x}_\star^\epsilon)\right), \quad \mu_\mathrm{L}^\epsilon = \max_{\mathbf{w}\in\Delta_m} \lambda_{\min}\left(\sum_{i=1}^m w_i \nabla^2 f_i(\widehat{\mathbf{x}_\star^\epsilon})\right).$$

Now, we show the formal theorem that for $\epsilon > 0$ PAMOO converges to $\widehat{\mathbf{x}_\star^\epsilon}$.

**Theorem E.3** ($\mu_\mathrm{L}^\epsilon$ Approximation Convergence of PAMOO)**.** *Suppose* $\{f_i\}_{i\in[m]}$ *are* $\beta$ *smooth,* $M_\mathrm{f}$ *self-concordant,* $\mathcal{C}_\epsilon$ *is an*

$\epsilon$*-ASS and that* $\mu_\mathrm{L}^\epsilon > 0$*, where* $\min\left\{\frac{\beta}{8}, \frac{1}{2\beta}\right\}\frac{\mu_\mathrm{L}^{\epsilon\,3}}{4^4\beta^4 mM_\mathrm{f}^2} \ge \epsilon > 0$*. Let* $k_0 := \left\lceil \frac{4^4\beta\left(\|\mathbf{x}_0 - \widehat{\mathbf{x}_\star^\epsilon}\|\frac{3\sqrt{m\beta}M_\mathrm{f}}{\sqrt{\mu_\mathrm{L}^\epsilon}} - 1\right)}{3\mu_\mathrm{L}^\epsilon} \right\rceil$ *where* $\|\cdot\|$ *is the*

*Euclidean-norm. Then,* Weighted-GD *instantiated with* PAMOO *weight-optimizer and* $\eta = 1$ *converges to* $\widehat{\mathbf{x}_\star^\epsilon}$ *with rate:*

$$\|\mathbf{x}_k - \widehat{\mathbf{x}_\star^\epsilon}\| \le \begin{cases} \|\mathbf{x}_{k_0} - \widehat{\mathbf{x}_\star^\epsilon}\|\left(1 - \frac{3\mu_\mathrm{L}^\epsilon}{32\beta}\right)^{\frac{k-k_0}{2}} + \sqrt{\frac{16\epsilon}{3\mu_\mathrm{L}^\epsilon} + \frac{32\sqrt{2\beta\epsilon}}{9\sqrt{\mu_\mathrm{L}}mM_\mathrm{f}}} & k \ge k_0 \\ \|\mathbf{x}_0 - \widehat{\mathbf{x}_\star^\epsilon}\| - k\frac{\mu_\mathrm{L}^{\epsilon\,3/2}}{64\beta^2\sqrt{m}M_\mathrm{f}} & o.w. \end{cases}$$

*Proof.* Combining Eq. (16) with the update rule of PAMOO, and plugging in $\widehat{\mathbf{x}_\star^\epsilon}$, i.e.

$$\mathbf{w}_k \in \underset{\mathbf{w} \in \mathbb{R}_+^m}{\arg\max} \, 2 \left( f_{\mathbf{w}_k}(\mathbf{x}_k) - f_{\mathbf{w}_k}(\widehat{\mathbf{x}_\star^\epsilon}) \right) - \|\nabla f_{\mathbf{w}_k}(\mathbf{x}_k)\|^2,$$

the following holds

$$\left\|\mathbf{x}_{k+1} - \widehat{\mathbf{x}_\star^\epsilon}\right\|^2 \leq \left\|\mathbf{x}_k - \widehat{\mathbf{x}_\star^\epsilon}\right\|^2 - \max_{\mathbf{w} \in \mathbb{R}_+^m} \left\{ 2 \left( f_{\mathbf{w}}(\mathbf{x}_k) - f_{\mathbf{w}}(\widehat{\mathbf{x}_\star^\epsilon}) \right) - \|\nabla f_{\mathbf{w}}(\mathbf{x}_k)\|^2 \right\} \tag{18}$$

Denote $\mathbf{w}_\star^\epsilon = \underset{\mathbf{w} \in \Delta_{m,w_{\min}}}{\arg\max} \, \lambda_{\min} \left( \sum_{i=1}^m w_i \nabla^2 f_i(\widehat{\mathbf{x}_\star^\epsilon}) \right)$, $a_k = f_{\mathbf{w}_\star^\epsilon}(\mathbf{x}_k) - f_{\mathbf{w}_\star^\epsilon}(\widehat{\mathbf{x}_\star^\epsilon})$, and $b_k = \left\|\nabla f_{\mathbf{w}_\star^\epsilon}(\mathbf{x}_k)\right\|^2$. Let $w(\mathbf{x}_k) = \mathbf{w}_\star^\epsilon \frac{a_k}{b_k} \in \mathbb{R}_+^m$, we can lower bound of the last expression as follows

$$\max_{\mathbf{w} \in \mathbb{R}_+^m} \left[ 2 \left( f_{\mathbf{w}}(\mathbf{x}_k) - f_{\mathbf{w}}(\widehat{\mathbf{x}_\star^\epsilon}) \right) - \|\nabla f_{\mathbf{w}}(\mathbf{x}_k)\|^2 \right] \geq 2 \left( f_{\mathbf{w}(\mathbf{x}_k)}(\mathbf{x}_k) - f_{\mathbf{w}(\mathbf{x}_k)}(\widehat{\mathbf{x}_\star^\epsilon}) \right) - \left\|\nabla f_{\mathbf{w}(\mathbf{x}_k)}(\mathbf{x}_k)\right\|^2$$

$$= \frac{\left( f_{\mathbf{w}_\star^\epsilon}(\mathbf{x}_k) - f_{\mathbf{w}_\star^\epsilon}(\widehat{\mathbf{x}_\star^\epsilon}) \right)^2}{\left\|\nabla f_{\mathbf{w}_\star^\epsilon}(\mathbf{x}_k)\right\|^2}.$$

Denote $\mathbf{x}_\star \in \arg\min_\mathbf{x} f_{\mathbf{w}_\star^\epsilon}(\mathbf{x})$. Then, it holds that

$$\frac{\left( f_{\mathbf{w}_\star^\epsilon}(\mathbf{x}_k) - f_{\mathbf{w}_\star^\epsilon}(\widehat{\mathbf{x}_\star^\epsilon}) \right)^2}{\left\|\nabla f_{\mathbf{w}_\star^\epsilon}(\mathbf{x}_k)\right\|^2}$$

$$= \frac{\left( f_{\mathbf{w}_\star^\epsilon}(\mathbf{x}_k) - f_{\mathbf{w}_\star^\epsilon}(\mathbf{x}_\star) + f_{\mathbf{w}_\star^\epsilon}(\mathbf{x}_\star) - f_{\mathbf{w}_\star^\epsilon}(\widehat{\mathbf{x}_\star^\epsilon}) \right) \left( f_{\mathbf{w}_\star^\epsilon}(\mathbf{x}_k) - f_{\mathbf{w}_\star^\epsilon}(\widehat{\mathbf{x}_\star^\epsilon}) \right)}{\left\|\nabla f_{\mathbf{w}_\star^\epsilon}(\mathbf{x}_k)\right\|^2}$$

$$= \frac{\left( f_{\mathbf{w}_\star^\epsilon}(\mathbf{x}_k) - f_{\mathbf{w}_\star^\epsilon}(\mathbf{x}_\star) \right) \left( f_{\mathbf{w}_\star^\epsilon}(\mathbf{x}_k) - f_{\mathbf{w}_\star^\epsilon}(\widehat{\mathbf{x}_\star^\epsilon}) \right)}{\left\|\nabla f_{\mathbf{w}_\star^\epsilon}(\mathbf{x}_k)\right\|^2} + \frac{\left( f_{\mathbf{w}_\star^\epsilon}(\mathbf{x}_\star) - f_{\mathbf{w}_\star^\epsilon}(\widehat{\mathbf{x}_\star^\epsilon}) \right) \left( f_{\mathbf{w}_\star^\epsilon}(\mathbf{x}_k) - f_{\mathbf{w}_\star^\epsilon}(\widehat{\mathbf{x}_\star^\epsilon}) \right)}{\left\|\nabla f_{\mathbf{w}_\star^\epsilon}(\mathbf{x}_k)\right\|^2}$$

$$= \frac{\left( f_{\mathbf{w}_\star^\epsilon}(\mathbf{x}_k) - f_{\mathbf{w}_\star^\epsilon}(\mathbf{x}_\star) \right) \left( f_{\mathbf{w}_\star^\epsilon}(\mathbf{x}_k) - f_{\mathbf{w}_\star^\epsilon}(\widehat{\mathbf{x}_\star^\epsilon}) \right)}{\left\|\nabla f_{\mathbf{w}_\star^\epsilon}(\mathbf{x}_k)\right\|^2} + \frac{\left( f_{\mathbf{w}_\star^\epsilon}(\mathbf{x}_\star) - f_{\mathbf{w}_\star^\epsilon}(\widehat{\mathbf{x}_\star^\epsilon}) \right) \left( f_{\mathbf{w}_\star^\epsilon}(\mathbf{x}_k) - f_{\mathbf{w}_\star^\epsilon}(\mathbf{x}_\star) \right)}{\left\|\nabla f_{\mathbf{w}_\star^\epsilon}(\mathbf{x}_k)\right\|^2}$$

$$+ \frac{\left( f_{\mathbf{w}_\star^\epsilon}(\mathbf{x}_\star) - f_{\mathbf{w}_\star^\epsilon}(\widehat{\mathbf{x}_\star^\epsilon}) \right)^2}{\left\|\nabla f_{\mathbf{w}_\star^\epsilon}(\mathbf{x}_k)\right\|^2}$$

$$\geq \frac{\left( f_{\mathbf{w}_\star^\epsilon}(\mathbf{x}_k) - f_{\mathbf{w}_\star^\epsilon}(\mathbf{x}_\star) \right) \left( f_{\mathbf{w}_\star^\epsilon}(\mathbf{x}_k) - f_{\mathbf{w}_\star^\epsilon}(\widehat{\mathbf{x}_\star^\epsilon}) \right)}{\left\|\nabla f_{\mathbf{w}_\star^\epsilon}(\mathbf{x}_k)\right\|^2} + \frac{\left( f_{\mathbf{w}_\star^\epsilon}(\mathbf{x}_\star) - f_{\mathbf{w}_\star^\epsilon}(\widehat{\mathbf{x}_\star^\epsilon}) \right) \left( f_{\mathbf{w}_\star^\epsilon}(\mathbf{x}_k) - f_{\mathbf{w}_\star^\epsilon}(\mathbf{x}_\star) \right)}{\left\|\nabla f_{\mathbf{w}_\star^\epsilon}(\mathbf{x}_k)\right\|^2}.$$

Since $\mathbf{w}_\star^\epsilon \in \Delta_{m,w_{\min}}$ it holds that $f_{\mathbf{w}_\star^\epsilon}$ is $\beta$ smooth. Then, it holds that $\left\|\nabla f_{\mathbf{w}_\star^\epsilon}(\mathbf{x})\right\|^2 \leq 2\beta \left( f_{\mathbf{w}_\star^\epsilon}(\mathbf{x}) - f_{\mathbf{w}_\star^\epsilon}(\mathbf{x}_\star) \right)$ for every $\mathbf{x}$, and we have

$$\max_{\mathbf{w} \in \mathbb{R}_+^m} \left[ 2 \left( f_{\mathbf{w}}(\mathbf{x}_k) - f_{\mathbf{w}}(\widehat{\mathbf{x}_\star^\epsilon}) \right) - \|\nabla f_{\mathbf{w}}(\mathbf{x}_k)\|^2 \right] \geq \frac{f_{\mathbf{w}_\star^\epsilon}(\mathbf{x}_k) - f_{\mathbf{w}_\star^\epsilon}(\widehat{\mathbf{x}_\star^\epsilon})}{2\beta} + \frac{f_{\mathbf{w}_\star^\epsilon}(\mathbf{x}_\star) - f_{\mathbf{w}_\star^\epsilon}(\widehat{\mathbf{x}_\star^\epsilon})}{2\beta}.$$

Plugging this in Eq. (18), we arrive to

$$\left\|\mathbf{x}_{k+1} - \widehat{\mathbf{x}_\star^\epsilon}\right\|^2 \leq \left\|\mathbf{x}_k - \widehat{\mathbf{x}_\star^\epsilon}\right\|^2 - \frac{f_{\mathbf{w}_\star^\epsilon}(\mathbf{x}_k) - f_{\mathbf{w}_\star^\epsilon}(\widehat{\mathbf{x}_\star^\epsilon})}{2\beta} + \frac{f_{\mathbf{w}_\star^\epsilon}(\widehat{\mathbf{x}_\star^\epsilon}) - f_{\mathbf{w}_\star^\epsilon}(\mathbf{x}_\star)}{2\beta}$$

$$\leq \left\|\mathbf{x}_k - \widehat{\mathbf{x}_\star^\epsilon}\right\|^2 - \frac{f_{\mathbf{w}_\star^\epsilon}(\mathbf{x}_k) - f_{\mathbf{w}_\star^\epsilon}(\widehat{\mathbf{x}_\star^\epsilon})}{2\beta} + \frac{\epsilon}{2\beta}.$$

By Lemma D.1 it holds that $f_{\mathbf{w}_\star}$ is $\widehat{M_f} := 1/\sqrt{w_{\min}} M_f$ self concordant. Then, From Lemma C.2, and its lower bound , i.e.

$\omega\left(t\right) \geq \frac{t^2}{2(1+t)}$, we have

$$f_{\mathbf{w}_\star^\epsilon}(\mathbf{x}_k) - f_{\mathbf{w}_\star^\epsilon}(\widehat{\mathbf{x}_\star^\epsilon}) \geq \langle \nabla f_{\mathbf{w}_\star^\epsilon}(\widehat{\mathbf{x}_\star^\epsilon}), \mathbf{x}_k - \widehat{\mathbf{x}_\star^\epsilon}\rangle + \frac{\left\|\mathbf{x}_k - \widehat{\mathbf{x}_\star^\epsilon}\right\|_{\widehat{\mathbf{x}_\star^\epsilon}, \mathbf{w}_\star^\epsilon}^2}{2\left(1 + \widehat{M}_\mathrm{f}\left\|\mathbf{x}_k - \widehat{\mathbf{x}_\star^\epsilon}\right\|_{\widehat{\mathbf{x}_\star^\epsilon}, \mathbf{w}_\star^\epsilon}\right)}$$

$$\geq -\left\|\nabla f_{\mathbf{w}_\star^\epsilon}(\widehat{\mathbf{x}_\star^\epsilon})\right\|\left\|\mathbf{x}_k - \widehat{\mathbf{x}_\star^\epsilon}\right\| + \frac{\left\|\mathbf{x}_k - \widehat{\mathbf{x}_\star^\epsilon}\right\|_{\widehat{\mathbf{x}_\star^\epsilon}, \mathbf{w}_\star^\epsilon}^2}{2\left(1 + \widehat{M}_\mathrm{f}\left\|\mathbf{x}_k - \widehat{\mathbf{x}_\star^\epsilon}\right\|_{\widehat{\mathbf{x}_\star^\epsilon}, \mathbf{w}_\star^\epsilon}\right)}.$$

By Lemma C.1, we have that $\left\|\nabla f_{\mathbf{w}_\star^\epsilon}(\widehat{\mathbf{x}_\star^\epsilon})\right\|^2 \leq 2\beta\left(f_{\mathbf{w}_\star^\epsilon}(\widehat{\mathbf{x}_\star^\epsilon}) - f_{\mathbf{w}_\star^\epsilon}(\mathbf{x}_\star)\right) \leq 2\beta\epsilon$, and, hence, $-\left\|\nabla f_{\mathbf{w}_\star^\epsilon}(\widehat{\mathbf{x}_\star^\epsilon})\right\| \geq -\sqrt{2\beta\epsilon}$. Plugging this back results with the following

$$\left\|\mathbf{x}_{k+1} - \widehat{\mathbf{x}_\star^\epsilon}\right\|^2 \leq \left\|\mathbf{x}_k - \widehat{\mathbf{x}_\star^\epsilon}\right\|^2 - \frac{\left\|\mathbf{x}_k - \widehat{\mathbf{x}_\star^\epsilon}\right\|_{\widehat{\mathbf{x}_\star^\epsilon}, \mathbf{w}_\star^\epsilon}^2}{4\beta\left(1 + \widehat{M}_\mathrm{f}\left\|\mathbf{x}_k - \widehat{\mathbf{x}_\star^\epsilon}\right\|_{\widehat{\mathbf{x}_\star^\epsilon}, \mathbf{w}_\star^\epsilon}\right)} + \frac{\epsilon}{2\beta} + \sqrt{2\beta\epsilon}\left\|\mathbf{x}_k - \widehat{\mathbf{x}_\star^\epsilon}\right\|$$

Note that $\lambda_{\min}\left(\nabla^2 f_{\mathbf{w}_\star^\epsilon}(\widehat{\mathbf{x}_\star^\epsilon})\right)\left\|\widehat{\mathbf{x}_\star^\epsilon} - \mathbf{x}_k\right\|^2 \leq \left\|\widehat{\mathbf{x}_\star^\epsilon} - \mathbf{x}_k\right\|_{\widehat{\mathbf{x}_\star^\epsilon}, \mathbf{w}_\star^\epsilon}^2 \leq \lambda_{\max}\left(\nabla^2 f_{\mathbf{w}_\star^\epsilon}(\widehat{\mathbf{x}_\star^\epsilon})\right)\left\|\widehat{\mathbf{x}_\star^\epsilon} - \mathbf{x}_k\right\|^2$. Since $\lambda_{\max}\left(\nabla^2 f_{\mathbf{w}_\star^\epsilon}(\widehat{\mathbf{x}_\star^\epsilon})\right) \leq \beta$ (smoothness), and since $w_{\min} = \mu_\mathrm{L}^\epsilon/(8m\beta)$, then $\widehat{M}_\mathrm{f} \leq 3\sqrt{m\beta}M_\mathrm{f}/\sqrt{\mu_\mathrm{L}^\epsilon}$. Thus, it holds that

$$\left\|\mathbf{x}_{k+1} - \widehat{\mathbf{x}_\star^\epsilon}\right\|^2 \leq \left\|\mathbf{x}_k - \widehat{\mathbf{x}_\star^\epsilon}\right\|^2 - \frac{1}{4\beta}\frac{\lambda_{\min}\left(\nabla^2 f_{\mathbf{w}_\star^\epsilon}(\widehat{\mathbf{x}_\star^\epsilon})\right)\left\|\widehat{\mathbf{x}_\star^\epsilon} - \mathbf{x}_k\right\|^2}{1 + \frac{3\sqrt{m}M_\mathrm{f}\beta}{\sqrt{\mu_\mathrm{L}^\epsilon}}\left\|\mathbf{x}_k - \widehat{\mathbf{x}_\star^\epsilon}\right\|} + \frac{\epsilon}{2\beta} + \sqrt{2\beta\epsilon}\left\|\mathbf{x}_k - \widehat{\mathbf{x}_\star^\epsilon}\right\|$$

Using the assumption that $\max_{\mathbf{w}\in\Delta_m} \lambda_{\min}\left(\nabla^2 f_{\mathbf{w}}(\widehat{\mathbf{x}^\epsilon})\right) = \mu_\mathrm{L}^\epsilon$, Lemma D.2, we have that

$$\lambda_{\min}\left(\nabla^2 f_{\mathbf{w}_\star^\epsilon}(\widehat{\mathbf{x}_\star^\epsilon})\right) \geq \max_{\mathbf{w}\in\Delta_m} \lambda_{\min}\left(\nabla^2 f_{\mathbf{w}}(\widehat{\mathbf{x}_\star^\epsilon})\right) - 2mw_{\min}\beta \geq \mu_\mathrm{L}^\epsilon - \mu_\mathrm{L}^\epsilon/4 = (3/4)\mu_\mathrm{L}^\epsilon.$$

Finally, we obtain the recurring equation we wish:

$$\left\|\mathbf{x}_{k+1} - \widehat{\mathbf{x}_\star^\epsilon}\right\|^2 \leq \left\|\mathbf{x}_k - \widehat{\mathbf{x}_\star^\epsilon}\right\|^2 - \frac{3\mu_\mathrm{L}^\epsilon}{16\beta}\frac{\left\|\widehat{\mathbf{x}_\star^\epsilon} - \mathbf{x}_k\right\|^2}{1 + \frac{3\sqrt{m}M_\mathrm{f}\beta}{\sqrt{\mu_\mathrm{L}^\epsilon}}\left\|\mathbf{x}_k - \widehat{\mathbf{x}_\star^\epsilon}\right\|} + \frac{\epsilon}{2\beta} + \sqrt{2\beta\epsilon}\left\|\mathbf{x}_k - \widehat{\mathbf{x}_\star^\epsilon}\right\|$$

Denote $\alpha_1 = \frac{3\mu_\mathrm{L}^\epsilon}{16\beta}$, $\alpha_2 = \frac{3\sqrt{m}\beta M_\mathrm{f}}{\sqrt{\mu_\mathrm{L}^\epsilon}}$, $\alpha_3 = \frac{\epsilon}{2\beta}$, and $\alpha_4 = \sqrt{2\beta\epsilon}$. Note that $\alpha_1 \in (0,1)$, $\alpha_2 \in \mathbb{R}_+$, $\alpha_3 \leq \frac{\alpha_1^2}{256\alpha_2^2}$, and $\alpha_4 \leq \frac{\alpha_1}{4\alpha_2}$. and $\left\|\mathbf{x} - \widehat{\mathbf{x}_\star^\epsilon}\right\| \in \mathbb{R}_+$ for every $\mathbf{x} \in \mathbb{R}^n$, we arrive to the recurrence relation analyzed in Lemma C.6. Then, we obtain

$$\left\|\mathbf{x}_k - \widehat{\mathbf{x}_\star^\epsilon}\right\| \leq \begin{cases} \left\|\mathbf{x}_{k_0} - \widehat{\mathbf{x}_\star^\epsilon}\right\|\left(1 - \frac{3\mu_\mathrm{L}^\epsilon}{32\beta}\right)^{\frac{k-k_0}{2}} + \sqrt{\frac{16\epsilon}{3\mu_\mathrm{L}^\epsilon} + \frac{32\sqrt{2\beta\epsilon}}{9\sqrt{\mu_\mathrm{L}m}M_\mathrm{f}}} & k \geq k_0 \\ \left\|\mathbf{x}_0 - \widehat{\mathbf{x}_\star^\epsilon}\right\| - k\frac{\mu_\mathrm{L}^{3/2}}{64\beta^2\sqrt{m}M_\mathrm{f}} & o.w. \end{cases}$$

where $k_0 := \left\lceil \frac{4^4\beta\left(\left\|\mathbf{x}_0 - \widehat{\mathbf{x}_\star^\epsilon}\right\|\frac{3\sqrt{m}\beta M_\mathrm{f}}{\sqrt{\mu_\mathrm{L}^\epsilon}} - 1\right)}{3\mu_\mathrm{L}^\epsilon} \right\rceil$. $\qquad\square$

## F. Missing Proofs

### F.1. Proof of Proposition 4.1

Let us restate the claim:

**Proposition 4.1** (Unique Optimal Solution). *Assume there exists $\mathbf{x}_\star \in \mathbb{R}^n$ that simultaneously minimizes $\{f_i\}_{i\in[m]}$, namely, solves Eq. (1). If $\max_{\mathbf{w}\in\Delta_m} \lambda_{\min}\left(\nabla^2 f_{\mathbf{w}}(\mathbf{x}_\star)\right) > 0$ then $\mathbf{x}_\star$ is unique.*

*Proof.* Let $\mathbf{x}_\star$ be a minimizer of all functions $\{f_i\}_{i \in [m]}$ which exists due to the AMOO assumption, namely, a solution of

$$\mathbf{x}_\star \in \arg \min_{\mathbf{x}} f_i(\mathbf{x}) \; \forall i \in [m]. \tag{19}$$

By assumption, it holds that for the weight vector $\mathbf{w}_\star \in \arg \max_{\mathbf{w} \in \Delta_m} \lambda_{\min} \left( \sum_{i=1}^m w_i \nabla^2 f_i(\mathbf{x}_\star) \right)$ it holds that $\lambda_{\min} \left( \sum_{i=1}^m \nabla^2 f_{\mathbf{w}_\star}(\mathbf{x}_\star) \right) > 0$, namely,

$$\nabla^2 f_{\mathbf{w}_\star}(\mathbf{x}_\star) \succ 0. \tag{20}$$

Notice that $\mathbf{x}_\star$ is a minimizer of $f_{\mathbf{w}_\star}$ (Lemma C.4), and that $f_{\mathbf{w}_\star}$ is a convex function, since $f_i$ are convex and $\mathbf{w}_\star$ has non-negative components. Combining with Eq. (20), it implies that $\mathbf{x}_\star$ is a unique minimizer of $f_{\mathbf{w}_\star}$.

Assume, by way of contradiction, there exists an additional minimizer that solves Eq. (19), denote by $\widehat{\mathbf{x}}_\star$. Since it is a solution of Eq. (19), it is also a minimizer of $f_{\mathbf{w}_\star}$. This contradicts the fact $f_{\mathbf{w}_\star}$ has a unique optimal solution $\mathbf{x}_\star$.

$\square$

### F.2. Proof of Proposition 4.2

The proof of Proposition 4.2 is a corollary of Theorem C.1 (Weyl's Theorem). We establish the result for a general deviation in Hessian matrices without requiring it to be necessarily diagonal.

Let us restate the result:

**Proposition 4.2.** *Assume that for all $i \in [m]$ and $\mathbf{x} \in \mathbb{R}^n$ $||\nabla^2 f_i(\mathbf{x}) - \mathrm{Diag}\left(\nabla^2 f_i(\mathbf{x})\right)||_2 \leq \|\Delta\|$ where $\|\mathbf{A}\|_2$ is the spectral norm of $\mathbf{A} \in \mathbb{R}^{n \times n}$. Let $\mathbf{w}_\star \in \arg \max_{\mathbf{w} \in \Delta_m} \lambda_{\min} \left( \sum_i w_i \nabla^2 \mathrm{Diag}\left(f_i(\mathbf{x})\right)\right)$. Then, $\lambda_{\min}\left(\sum_i w_{\star,i} \nabla^2 f_i(\mathbf{x})\right) \geq \mu_G - 2\|\Delta\|$.*

*Proof.* Denote $\mathbf{A}_i := \nabla^2 f_i(\mathbf{x}) + \Delta_i$ for every $i \in [m]$, and $\sum_i^m \Delta_i = \Delta$. Let $\mathbf{w}_\star$, and $\hat{\mathbf{w}}_\star$ denote the solution of,

$$\mathbf{w}_\star \in \arg \max_{\mathbf{w} \in \Delta} \lambda_{\min} \left( \sum_i w_i \nabla^2 f_i(\mathbf{x}) \right), \quad \text{and} \quad \hat{\mathbf{w}}_\star \in \arg \max_{\mathbf{w} \in \Delta} \lambda_{\min} \left( \sum_i w_i \mathbf{A}_i \right),$$

respectively. Let $g(\mathbf{w}_\star)$, and $\hat{g}(\hat{\mathbf{w}}_\star)$ denote the optimal value, $g(\mathbf{w}_\star) = \lambda_{\min}\left(\sum_i w_{\star,i} \nabla^2 f_i(\mathbf{x})\right)$, and $\hat{g}(\hat{\mathbf{w}}_\star) = \lambda_{\min}\left(\sum_i \hat{w}_{\star,i} \mathbf{A}_i\right)$. Then, since $\hat{g}(\mathbf{w}_\star) - \hat{g}(\hat{\mathbf{w}}_\star) \leq 0$ by the optimality of $\hat{\mathbf{w}}_\star$ on $\hat{g}$, the following holds

$$g(\mathbf{w}_\star) = g(\mathbf{w}_\star) - \hat{g}(\mathbf{w}_\star) + \hat{g}(\mathbf{w}_\star) - \hat{g}(\hat{\mathbf{w}}_\star) + \hat{g}(\hat{\mathbf{w}}_\star) - g(\hat{\mathbf{w}}_\star) + g(\hat{\mathbf{w}}_\star)$$
$$\leq |g(\mathbf{w}_\star) - \hat{g}(\mathbf{w}_\star)| + |\hat{g}(\hat{\mathbf{w}}_\star) - g(\hat{\mathbf{w}}_\star)| + g(\hat{\mathbf{w}}_\star)$$

Using Theorem C.1 (Weyl's Theorem) the followings are hold: $|g(\mathbf{w}_\star) - \hat{g}(\mathbf{w}_\star)| \leq \Delta$, and $|\hat{g}(\hat{\mathbf{w}}_\star) - g(\hat{\mathbf{w}}_\star)| \leq \Delta$. Then, we obtain

$$g(\mathbf{w}_\star) \leq 2\|\Delta\| + g(\hat{\mathbf{w}}_\star)$$

Finally, since $g(\mathbf{w}_\star) \geq \mu_G$, by Definition 4.1, we obtain the proof. $\square$

### F.3. Proof of Lemma C.5

Let us restate the claim:

**Lemma C.5** (Recurrence bound AMOO)**.** *Let $\{r_k\}_{k \geq 0}$ be a sequence of non-negative real numbers that satisfy the recurrence relation*

$$r_{k+1}^2 \leq r_k^2 - \alpha_1 \frac{r_k^2}{1 + \alpha_2 r_k},$$

*where $\alpha_1 \in [0, 2)$ and $\alpha_2 \in \mathbb{R}_+$. Let $k_0 := \left\lceil \frac{4(r_0 \alpha_2 - 1)}{\alpha_1} \right\rceil$. Then, $r_k$ is bounded by*

$$r_k \leq \begin{cases} r_{k_0} \left(1 - \frac{\alpha_1}{2}\right)^{\frac{k - k_0}{2}} & k \geq k_0 \\ r_0 - \frac{\alpha_1}{4\alpha_2} k & o.w. \end{cases}.$$

*Proof.* We split the proof into two regimes, the incremental convergence and linear convergence regime.

**Incremental convergence, $r_k > 1/\alpha_2$.** With this assumption we have $1 + r_k\alpha_2 < 2r_k\alpha_2$. Then, the following bound holds:

$$r_{k+1} \leq r_k\sqrt{1 - \frac{\alpha_1}{2\alpha_2 r_k}}.$$

Recall that $\sqrt{1-y} \leq 1 - \frac{y}{2}$ for every $y \leq 1$. Since $\frac{1}{\alpha_2 r_k} < 1$ we have $\frac{\alpha_1}{2\alpha_2 r_k} < \frac{\alpha_1}{2} < 1$. Hence,

$$r_{k+1} \leq r_k\left(1 - \frac{\alpha_1}{4\alpha_2 r_k}\right) = r_k - \frac{\alpha_1}{4\alpha_2}.$$

For every $k' < k$ the recursive equation is still in the incremental convergence regime. Thus, for every $k' < k$ holds that $r_{k'} > 1/\alpha_2$.

By solving $1/\alpha_2 \geq r_0 - k_0\alpha_1/4\alpha_2$ we conclude the maximal iteration after which $r_k \leq 1/\alpha_2$, namely, after at most $k_0$ iterates $r_k$ out from the incremental convergence regime.

**Linear convergence, $r_k \leq 1/\alpha_2$.** With this assumption we have the following bound on the recursive equation:

$$r_{k+1}^2 \leq \left(1 - \frac{\alpha_1}{2}\right)r_k^2.$$

Further, since for every $k' \geq k$ it holds that $r_{k'} \leq r_k \leq 1/\alpha_2$ the recursive equation continues in the linear convergence regime. Thus, after at most $k_0$ iterations $r_k$ is in the linear convergence regime.

$\square$

### F.4. Proof of Lemma C.6

Let us restate the claim:

**Lemma C.6** (Recurrence bound $\epsilon$-AAMOO). *Let $\{r_k\}_{k\geq 0}$ be a sequence of non-negative real numbers that satisfy the recurrence relation*

$$r_{k+1}^2 \leq r_k^2 - \alpha_1\frac{r_k^2}{1 + \alpha_2 r_k} + \alpha_3 + \alpha_4 r_k,$$

*where $\alpha_1 \in (0,2)$, $\alpha_2 \in \mathbb{R}_+$, $\alpha_3 \leq \frac{\alpha_1^2}{256\alpha_2^2}$, and $\alpha_4 \leq \frac{\alpha_1}{4\alpha_2}$. Let $k_0 := \left\lceil \frac{16(r_0\alpha_2 - 1)}{\alpha_1} \right\rceil$. Then, $r_k$ is bounded by*

$$r_k \leq \begin{cases} r_{k_0}\left(1 - \frac{\alpha_1}{2}\right)^{\frac{k'-k}{2}} + \sqrt{\frac{2\alpha_3}{\alpha_1} + \frac{2\alpha_4}{\alpha_1\alpha_2}} & k \geq k_0 \\ r_0 - \frac{\alpha_1}{16\alpha_2}k & o.w. \end{cases}.$$

*Proof.* We split the proof into two regimes, the incremental convergence and linear convergence regime.

**Incremental convergence, $r_k > 1/\alpha_2$.** With this assumption we have $1 + r_k\alpha_2 < 2r_k\alpha_2$. Then, the following bound holds:

$$r_{k+1} \leq \sqrt{r_k^2 - \frac{\alpha_1 r_k}{2\alpha_2} + \alpha_4 r_k + \alpha_3} = \sqrt{r_k^2\left(1 - \frac{\alpha_1 - 2\alpha_2\alpha_4}{2\alpha_2 r_k}\right) + \alpha_3} \leq r_k\sqrt{1 - \frac{\alpha_1}{4\alpha_2 r_k}} + \sqrt{\alpha_3}.$$

The third relation holds since $\alpha_4 \leq \frac{\alpha_1}{4\alpha_2}$ by assumption which implies $\alpha_1/2 \geq 2\alpha_2\alpha_4$ and by $\sqrt{a+b} \leq \sqrt{a} + \sqrt{b}$ for $a, b \geq 0$.

Recall that $\sqrt{1-y} \leq 1 - y/2$ for every $y \leq 1$. Since $\frac{1}{\alpha_2 r_k} < 1$ we have $\frac{\alpha_1}{2\alpha_2 r_k} < \frac{\alpha_1}{2} < 1$. Hence,

$$r_{k+1} \leq r_k\left(1 - \frac{\alpha_1}{8\alpha_2 r_k}\right) + \sqrt{\alpha_3} = r_k - \frac{\alpha_1}{8\alpha_2} + \sqrt{\alpha_3} \leq r_k - \frac{\alpha_1}{16\alpha_2},$$

since $\alpha_3 \leq \frac{\alpha_1^2}{256\alpha_2^2} = \frac{\alpha_1^2}{16^2\alpha_2^2}$ by assumption. For every $k' < k$ the recursive equation is still in the incremental convergence regime. Thus, for every $k' < k$ holds that $r_{k'} > 1/\alpha_2$.

By solving $1/\alpha_2 \geq r_0 - k_0\alpha_1/16\alpha_2$ we conclude the maximal iteration after which $r_k \leq 1/\alpha_2$, namely, after at most $k_0$ iterates $r_k$ out from the incremental convergence regime.

**Linear convergence, $r_k \leq 1/\alpha_2$.** With this assumption we have the following bound on the recursive equation:

$$r_{k+1}^2 \leq \left(1 - \frac{\alpha_1}{2}\right)r_k^2 + \alpha_3 + \alpha_4/\alpha_2 = \left(1 - \frac{\alpha_1}{2}\right)r_k^2 + \alpha', \tag{21}$$

where $\alpha' := \alpha_3 + \alpha_4/\alpha_2$. We will first show that $r_{k'} \leq 1/\alpha_2$ for all $k' \geq k$. Observe that

$$r_{k+1}^2 \leq \left(1 - \frac{\alpha_1}{2}\right)\frac{1}{\alpha_2^2} + \alpha' \leq \frac{1}{\alpha_2^2}$$

since $\alpha' = \alpha_3 + \frac{\alpha_4}{\alpha_2} \leq \frac{\alpha_1}{2\alpha_2^2}$ by assumption and by the fact $\alpha_1 \in (0,2)$. Hence, $r_{k+1} \leq \frac{1}{\alpha_2}$ which inductively implies that $r_{k'} \leq r_k \leq 1/\alpha_2$ for all $k' \geq k \geq k_0$.

Since in this regime, for all $k' \geq k$ Eq. (21) holds, we can upper bound the recursive relation by

$$r_{k'}^2 \leq \left(1 - \frac{\alpha_1}{2}\right)^{k'-k}r_k^2 + \sum_{t=0}^{\infty}\left(1 - \frac{\alpha_1}{2}\right)^t \alpha' \leq r_{k_0}^2\left(1 - \frac{\alpha_1}{2}\right)^{k'-k} + \frac{2\alpha'}{\alpha_1}$$

where the last inequality holds by summing the geometric series and since $1 - \alpha_1/2 \in (0,1)$. This inequality implies the result since

$$r_{k'} \leq r_{k_0}\left(1 - \frac{\alpha_1}{2}\right)^{\frac{k'-k}{2}} + \sqrt{\frac{2\alpha'}{\alpha_1}} = r_{k_0}\left(1 - \frac{\alpha_1}{2}\right)^{\frac{k'-k}{2}} + \sqrt{\frac{2\alpha_3}{\alpha_1} + \frac{2\alpha_4}{\alpha_1\alpha_2}}$$

by $\sqrt{a+b} \leq \sqrt{a} + \sqrt{b}$ for $a, b \geq 0$. Finally, since for every $k' \geq k$ it holds that $r_{k'} \leq r_k \leq 1/\alpha_2$ the recursive equation continues in the linear convergence regime. Thus, after at most $k_0$ iterations $r_k$ is in the linear convergence regime.

$\square$

### F.5. Proof of Lemma D.2

Let us restate the claim:

**Lemma D.2** (Continuity of Minimal Eigenvalue of Hessian). *Let $\mathbf{x} \in \mathbb{R}^n$. Further, let $\mathbf{w}_\star \in \arg\max_{\mathbf{w} \in \Delta_m} \lambda_{\min}\left(\nabla^2 f_{\mathbf{w}}(\mathbf{x})\right)$, $\widehat{\mathbf{w}} \in \arg\max_{\mathbf{w} \in \Delta_{m,w_{\min}}} \lambda_{\min}\left(\nabla^2 f_{\mathbf{w}}(\mathbf{x})\right)$. It holds that $\lambda_{\min}\left(\nabla^2 f_{\widehat{\mathbf{w}}}(\mathbf{x})\right) \geq \lambda_{\min}\left(\nabla^2 f_{\mathbf{w}_\star}(\mathbf{x})\right) - 2mw_{\min}\beta$.*

*Proof.* To establish the lemma we want to show that for any $\mathbf{w} \in \Delta_m$ there exists $\widehat{\mathbf{w}} \in \Delta_{m,w_{\min}}$ such that $\lambda_{\min}\left(\sum_i \hat{w}_i \nabla^2 f_i(\mathbf{x})\right) \geq \lambda_{\min}\left(\sum_i w_i \nabla^2 f_i(\mathbf{x})\right) - 2mw_{\min}\beta$. We start by bounding the following term $\left\|\nabla^2 f_{\mathbf{w}}(\mathbf{x}) - \nabla^2 f_{\hat{\mathbf{w}}}(\mathbf{x})\right\|_2$ for any $\mathbf{x} \in \mathbb{R}^n$. By the triangle inequality and the positive homogeneity, we have

$$\left\|\sum_i (w_i - \hat{w}_i)\nabla^2 f_i(\mathbf{x})\right\|_2 \leq \sum_i \left\|(w_i - \hat{w}_i)\nabla^2 f_i(\mathbf{x})\right\|_2 = \sum_i |w_i - \hat{w}_i| \left\|\nabla^2 f_i(\mathbf{x})\right\|_2 \leq \beta \sum_i |w_i - \hat{w}_i|,$$

while the last inequality holds since $\{f_i\}_{i \in [m]}$ are $\beta$ smooth. Since for any $\mathbf{w} \in \Delta_m$ there exist $\hat{\mathbf{w}} \in \Delta_{m,w_{\min}}$ such that $\sum_i |w_i - \hat{w}_i| \leq 2mw_{\min}$, we obtain that for every $\mathbf{x} \in \mathbb{R}^n$ it holds that

$$\left\|\nabla^2 f_{\mathbf{w}}(\mathbf{x}) - \nabla^2 f_{\hat{\mathbf{w}}}(\mathbf{x})\right\|_2 = \left\|\sum_i (w_i - \hat{w}_i)\nabla^2 f_i(\mathbf{x})\right\|_2 \leq 2mw_{\min}\beta.$$

Thus, by using Theorem C.1 we obtain that for any $\mathbf{w} \in \Delta_m$

$$|\lambda_{\min}(\nabla^2 f_{\mathbf{w}}(\mathbf{x})) - \lambda_{\min}(\nabla^2 f_{\hat{\mathbf{w}}}(\mathbf{x}))| \leq \left\|\nabla^2 f_{\mathbf{w}}(\mathbf{x}) - \nabla^2 f_{\hat{\mathbf{w}}}(\mathbf{x})\right\|_2 \leq 2mw_{\min}\beta.$$

By setting $\mathbf{w}$ as $\mathbf{w}_\star$ we conclude the result. $\square$

