# OpenReview forum: "Aligned Multi Objective Optimization"
_ICML.cc/2025/Conference — ICML 2025 poster_

### Official Review · Reviewer_54gT · 2025-03-10

**Overall Recommendation:** 2

**Summary:**

This paper focuses on multi-objective optimization (MOO). Differently from the typical MOO studies that focus on dealing with conflicts among objectives, this paper considers a so-called aligned MOO settings, where objectives align well with each other and no conflict occurs. This paper wants to develop methods that are able to improve aligned MOO from an optimization perspective. More specifically, the authors assume all objectives have the same optimal solution. In the meanwhile, three toy examples are provided to illustrate that a careful weight selection should be provided to achieve faster convergence as well as better scalability. As motivated by these findings, the authors first propose CAMOO by exploiting the so-called global adaptive strong convexity (it also guarantees the unique solution that minimizes all objectives). A convergence analysis is provided under a self-concordant assumption. Secondly, the authors  follow the similar idea from CAMOO, but choose the weight vector based on Polyak stepsizes, and propose PAMOO, which requires only gradients rather than Hessians as in CAMOO. The analysis is provided given the local strong convexity assumption (made at x* rather than any x).

**Claims And Evidence:**

Yes, most claims are clear to me.

**Essential References Not Discussed:**

No key references are missing, but some discussions on related works with convergence analysis of MOO should be provided. I suggest three important ones in the 'Other Strengths And Weaknesses'

**Experimental Designs Or Analyses:**

There are not enough experiments to validate the proposed algorithms. Only synthetic datasets are provided. More large-scale datasets and models need to be tested. In addition, the efficiency of the methods should be analyzed.

**Methods And Evaluation Criteria:**

There are not enough experiments to validate the proposed algorithms. Only synthetic datasets are provided.

**Other Comments Or Suggestions:**

See 'Other Strengths And Weaknesses'.

**Other Strengths And Weaknesses:**

Strength:

1.	Aligned MOO has not been well studied and understood, the authors start this problem from an interesting optimization perspective. The toy examples present the high-level ideas, which are easy to understand.

2.	The proposed methods that utilize either global or local strong convexity (for weighted average of objectives) are interesting and the linear convergence result makes senses. The analysis seems to be nontrivial by developing a inequality that follows from the self-concordance property.

3.	Two practical versions are provided for large-scale applications.

Weakness:

1.	Although studying aligned cases is important, the problems studied in this paper seems to oversimplify the problem. For example, the assumption that all objectives share the same unique minimizer might be hard to hold in practice. Perhaps some experiments could be provided to support this assumption. In addition, the global adaptive strong convexity requires to hold for any x and requires the existence of weights to ensure the strong convexity. This is also very restrictive. The self-concordant condition is also strong. In other words, although the theoretical results are encouraging (linear convergence without dependence on objective number), the results seem to be more natural under these additional assumptions.

2.	Following the previous point, I want to understand if the results hold, that says, when the assumptions are partially satisfied? For example, we can assumption only a subset of objectives satisfy these conditions, whereas for others, conflict could happen. Or the authors can think about more practical settings, where the assumptions could be justified.

3.	CAMOO is not practical. Computing the Hessian matrices and their smallest eigenvalue in each iteration can introduce non-trivial computional and momery cost. Thus, to me, PAMOO makes more sense as it uses only gradients. However, for PAMOO, solving argmax over w in each iteration K is not efficient, due to the O(K) cost in computing and storing the Jacobians (matrix of individual gradients). Although Section 5.1 discusses the practical implementation, e.g., running several steps of PGD, I am wondering how many steps to choose? If a similar convergence analysis can be developed for this practical version?

4.	The experiments are not convincing. Only simple synthetic experiments are provided. More experiments on large-scale MTL or MOO applications and real-world datasets should be provided.

5.	The paper focuses on GD. I wonder can the analysis be extended to stochastic settings. The following three important papers on the convergence  analysis of stochastic MOO could be helpful. I suggest a discussion could be provided.

(1)	The stochastic multi-gradient algorithm for multi-objective  optimization and its application to supervised machine learning

(2)	Mitigating gradient bias in multi-objective learning: A provably convergent approach.

(3)	Direction-oriented Multi-objective Learning: Simple and Provable Stochastic Algorithms

**Questions For Authors:**

See 'Other Strengths And Weaknesses'.

**Relation To Broader Scientific Literature:**

Study aligned MOO rather than previous works on MOO with conflicts.

**Theoretical Claims:**

I checked some proofs, and they look good to me.

---

> ### Author Rebuttal · Authors · 2025-03-31
>
> Thank you for the feedback and recognizing the positive aspects of our work. We appreciate the detailed and useful feedback. We will address the questions in their chronological order.
>
> 1. In Section 6 – where we generalize AMOO to approximate AMOO – we are not assuming a unique minimizer for all objectives. In addition, the linear convergence without dependence on the number of objectives is not natural; for example, if one chooses the weights to be uniform (1/m), the linear convergence depends on the number of objectives. Since this wasn't clear, we will add a section in the appendix that demonstrates this. Also, see section B in the appendix.
> Further, we believe that this work offers a new and rigorous perspective on the multi-task setting that becomes increasingly important in machine learning applications (e.g., in LLMs and RL research practitioners often use multiple reward functions or datasets).
>
> 2. It is a good question, thanks for raising it. Yes, in Section 6 we provide results for the setting in which functions are only approximately aligned, and show CAMOO and PAMOO both converge to an approximately optimal point for all objectives. The additional experiments we attached to the rebuttal corroborate these theoretical findings (see this anonymous link https://docs.google.com/document/d/e/2PACX-1vRbT_IuVcnR-RHPy95d9sIuiHa07DUmjA57HvWKjjWFUjST5udLeTUygnpnQ3AvSmiuc_y-r4JpiGYI/pub).
>
> 3. We agree that CAMOO is less practical compared to PAMOO. However, such an algorithm is very natural for the AMOO setting, given the intuition we provide in Figure 1, and gives useful intuition that may help deriving algorithmic improvements. Additionally, as the open source implementation we shared showcases, it is possible to make use of modern approaches for Hessian estimation (e.g., Hutchinson method) for implementing such an approach.
>
> 4. Our work focused on foundational work: formulating the AMOO setting, developing intuition and providing new algorithms with provable guarantees for the setting, as well as showing these algorithms are robust to the alignment assumption. The code we share with the research community showcases the feasibility of implementing the algorithms. We believe the next step of our work is conducting large-scale experiments in different domains (e.g., LLM, RL and computer vision) while also developing the theoretical foundations of the AMOO setting.
>
> 5. We also believe the stochastic setting is of interest and importance. Due to the significant contributions this work has put forward (see response to question 4) we believe such extension is proper for future work. We will expand the existing discussion on such a future extension further (see the Conclusions section) to highlight the importance of this direction. Thank you for raising this point.

---

### Official Review · Reviewer_ZyHT · 2025-03-14

**Overall Recommendation:** 4

**Summary:**

The paper introduces Aligned Multi-Objective Optimization (AMOO), addressing scenarios where multiple objectives share a common solution. Traditional MOO focuses on conflicting objectives, but here, aligned objectives enable simultaneous optimization. The authors propose CAMOO (curvature-aware weighting) and PAMOO (Polyak step-size-inspired weighting), both adaptively combining gradients. Theoretical guarantees show improved convergence rates over naive methods, with rates depending on problem-specific curvature parameters.

**Claims And Evidence:**

Claims are supported by theory (Theorems 4.2, 5.2) and experiments. The convergence analysis assumes self-concordance and smoothness, with proofs outlined in appendices. And Theorem 6.2 addresses approximate alignment.

**Essential References Not Discussed:**

No.

**Experimental Designs Or Analyses:**

Experiments validate CAMOO/PAMOO on synthetic tasks. Missing are tests on standard multi-task benchmarks and non-convex settings.

**Methods And Evaluation Criteria:**

Methods are sensible: CAMOO uses Hessian-based curvature, PAMOO leverages Polyak steps. Evaluation uses synthetic alignment scenarios (specification, selection, curvature) and a neural network matching task. Maybe broader benchmarks (e.g., standard multi-task datasets) can help to assess its practical impact.

**Other Comments Or Suggestions:**

The local curvature example (Figure 1) could benefit from clearer visualization.

**Other Strengths And Weaknesses:**

N/A

**Questions For Authors:**

For PAMOO, how does inaccuracy in $f_i(x_*)$ affect performance?

**Relation To Broader Scientific Literature:**

Connects well with multi-task learning (MTL) and MOO literature (e.g., MGDA, PCGrad), differentiating via aligned objectives.

**Theoretical Claims:**

Yes.

---

> ### Author Rebuttal · Authors · 2025-03-31
>
> Thank you for your interest in this work and the questions you raised! We will address them in their chronological order.
>
> 1. We will make the local curvature example figure clearer – thank you for pointing this out!
>
> 2. We provided results on inaccuracy of $f_i(x_{\star})$ in Section 6, where we assume the multi-objectives are only approximately aligned (we derived two results, one for the PAMOO and one for CAMOO algorithms). We conducted additional experiments, for providing additional empirical evidence, for the approximate AMOO to be included in the final version of the paper (see this anonymous link https://docs.google.com/document/d/e/2PACX-1vRbT_IuVcnR-RHPy95d9sIuiHa07DUmjA57HvWKjjWFUjST5udLeTUygnpnQ3AvSmiuc_y-r4JpiGYI/pub). Lastly, we note several prior works explored the question on designing gradient-descent based methods in which the learning rate is chosen via a Polyak method and the optimal value of a function is estimated adaptively (Hazan and Kakade 2019, Gower et al., 2021; Orvieto et al., 2022). We believe that generalizing these to the AMOO setting is a fertile ground for future research work, which we also mentioned in the main text.

---

### Official Review · Reviewer_YY6M · 2025-03-17

**Overall Recommendation:** 3

**Summary:**

This paper studies an interesting problem, i.e., aligned multi-objective optimization (MOO) where different objectives could share the same optimal solution. The paper introduces new algorithms for this setting, and provides theoretical convergence guarantees of the new algorithm under convexity assumptions of the objectives.

## update after rebuttal

I increase my score since the authors have addressed most of my concerns. A few suggestions are listed below.

1. It is strongly suggested that the authors include the run time comparison of different algorithms discussed in the rebuttal.
2. For the discussion on the extension to the nonconvex setting, the key concern I have is that the faster convergence rate of the proposed method compared to fixed weighting is achieved only in the convex setting. In the nonconvex setting, which is more common in modern machine learning, the proposed method generally no longer achieves a faster convergence rate compared to fixed weighting. This limits its practical use. More discussions should be provided.

**Claims And Evidence:**

1. The authors claim that aligned MOO is common for machine learning in modern settings, but they provide a theoretical analysis of convex objectives without connecting the assumptions to some modern machine learning examples.

**Essential References Not Discussed:**

There are several references that are not accurately referenced or properly discussed.

### MGDA related works
The discussion of MGDA related works in Section 2 is not accurate. The first works on MGDA is [1] by Fliege et al., followed by some other works in the stochastic settings, e.g., [2-4].

[1] Jorg Fliege and Benar Fux Svaiter. "Steepest descent methods for multicriteria optimization,’’ Mathematical methods of operations research, 2000.

[2] Suyun Liu and Luis Nunes Vicente. "The stochastic multi-gradient algorithm for multi-objective optimization and its application to supervised machine learning,’’ Annals of Operations Research, 2021.

[3] Shiji Zhou, Wenpeng Zhang, Jiyan Jiang, Wenliang Zhong, Jinjie Gu, and Wenwu Zhu, "On the convergence of stochastic multi-objective gradient manipulation and beyond,’’ NeurIPS, 2022.

[4] Lisha Chen, Heshan Fernando, Yiming Ying, Tianyi Chen. "Three-way trade-off in multi-objective learning: Optimization, generalization and conflict-avoidance,” NeurIPS, 2023.

### Fast algorithms for MOO
There are some other algorithms for fast implementation of MOO, which should be discussed and compared. See below.

[5] Ozan Sener, Vladlen Koltun, "Multi-Task Learning as Multi-Objective Optimization," NeurIPS 2018.

[6] Bo Liu, Yihao Feng, Peter Stone, and Qiang Liu. "Famo: Fast adaptive multitask optimization,” NeurIPS, 2023.

**Experimental Designs Or Analyses:**

No significant issues.

One minor issue is that the experiments are rather toy, and it is unclear how the experiments connect with modern machine learning in the introduction.

**Methods And Evaluation Criteria:**

Yes.

**Other Comments Or Suggestions:**

No.

**Other Strengths And Weaknesses:**

### Strengths

1. The studied problem is very interesting in the aligned MOO setting.
2. Theoretical results and empirical evaluations are provided.

### Weaknesses

1. The analyzed settings and the experiment settings are rather simple (convex objectives). It is unclear how they relate to modern machine learning. Can the assumptions made in the theoretical results be justified in some modern machine learning examples?

2. The descriptions of the theoretical results lack clarity and can be improved. See details in Questions.

**Questions For Authors:**

### Major

1. It would be better if the authors first provide a formal definition of aligned MOO, and how it connects to existing works. For example, does it mean that the Pareto front reduces to a single point?
2. On page 3, the authors discuss the linear convergence rate in $\Omega(\Delta)$ for example (i). This is confusing to me. As far as I know, $\Delta$ does not affect the convergence rate depending on the number of iterations, but it will affect the constant ratio in the convergence or optimization error bound. The discussion of convergence in $\Omega(\Delta)$ should be clearly defined and explained.


### Minor

1. Where is $m$ in Table 1?
2. In Theorem 6.2, what is "self-concordant"? Please move the definition in the appendix to the main text.
3. On Page 3, second column, "Existing algorithms suggest alternatives for choosing w", could you add some references to support this and add discussions on which paradigm you use?

I will consider raising my score if the questions and weaknesses above are properly addressed.

**Relation To Broader Scientific Literature:**

Compared to existing MOO works, the paper considers aligned MOO settings rather than the Pareto trade-off settings.

Compared to multi-fidelity optimization, this paper considers gradient-based optimization.

**Theoretical Claims:**

I mainly checked Appendix B and C and found no errors. Other results also seem reasonable.

---

> ### Author Rebuttal · Authors · 2025-03-31
>
> Thank you for the feedback and recognizing the positive aspects of our work. We appreciate the detailed and useful feedback.
>
> We wish to mention additional contributions of this work, beyond those that were mentioned in the review:
>
> 1. In Section 6 we study the approximate AMOO framework. There we showed that both CAMOO and PAMOO are robust to violations of the alignment assumptions (see this anonymous link https://docs.google.com/document/d/e/2PACX-1vRbT_IuVcnR-RHPy95d9sIuiHa07DUmjA57HvWKjjWFUjST5udLeTUygnpnQ3AvSmiuc_y-r4JpiGYI/pub).
>
> 2. The implementation of the CAMOO and PAMOO algorithms (see the supplementary material) showcases the feasibility of both algorithms. Additionally, our experiments go beyond the theoretical results when using neural networks with SGD or ADAM optimizers. This implementation allows future researchers to build upon our work and provides evidence for the usefulness of CAMOO and, even more so, of PAMOO, as the empirical results show.
>
> We will address the comments in their chronological order.
>
> References.
> Thank you for sharing the references. We will include them in the final draft (we also point out that reference [5] is included in the current draft and mentioned in Section 2).
>
> Weakness (1).
> 1. We agree that modern machine learning applications are not well described by a convex optimization framework. Nevertheless, the design of all modern optimization algorithms – SGD, ADAM and many more – is motivated by algorithms from the convex optimization framework. For example, ADAM is intimately related to the AdaGrad algorithm, which is designed for the convex setting (see https://arxiv.org/abs/1412.6980). Similarly, we believe that establishing the correctness of optimizers in the convex optimization framework is useful towards designing practical algorithms.
>
> Weakness Major.
> 1. Yes, you are correct and we will add this clarification to the paper. Since there is no tradeoff – by assumption all objectives can be simultaneously optimized – the Pareto front is a point. We provided a formal description of the general setting in Section (3), as well as a more nuanced description in Section (4) (e.g., Proposition (1) is an additional structural result about the setting).
> 2. In the specification example, the functions are strongly convex and smooth; for this case, the convergence of GD is (1-\Delta)^k. We will use this precise description in the text. Thank you!
>
> Minor.
> 1. This is one of the interesting findings in the paper in our opinion. The convergence rate does not dependent on $m$, the number of objectives. We will add clarification about this in the caption of Table (1) as it seems it is not properly conveyed to the reader.
> 2. We provided a formal definition in the Appendix. We will move it to the main text for improved clarity.
> 3. Most practical algorithms in multi-task learning specify different ways for choosing weights to output weighted loss (e.g., "Multi-task learning as multi-objective optimization" by Sener et al., "Conflict-averse gradient descent for multi-task learning" by Liu et al., and "Multi-Task Learning Using Uncertainty to Weigh Losses for Scene Geometry and Semantics" by Kendall et al.).

---

> > ### Comment · Reviewer_YY6M · 2025-04-07
> >
> > Thanks for the rebuttal. It addressed some of my concerns. Overall I think the studied problem is very interesting and provides some new perspective. However, the analysis is limited to convex problems, and the experiments are limited to small-scale problems, making it unclear whether the proposed method is useful in practice for modern machine learning with very large models. I check some of the proofs, update my reviews, and increase my score. I have a few more questions/suggestions:
> >
> > 1. I am not convinced by the claim that the proposed method can be faster in practice than the simplest weighted sum/linear scalarization method using a fixed weight. Though it has been proved that the required iterations for convergence are smaller for the proposed method, the method has much higher per-iteration complexity to solve for the best weight compared to using a fixed weight. It would be better if the authors could discuss the per-iteration complexity, and compare the total training time and computation complexity (per-iteration complexity times number of iterations) till convergence for different methods.
> >
> > 2. The theory is established for the convex setting. It would be better if the author could provide more discussion on how the theory could be extended to some modern problems with possibly nonconvex objectives.

---

> > > ### Author Response · Authors · 2025-04-08
> > >
> > > Thank you for these questions! As we detail below, we added an experiment to measure the running time of CAMOO, PAMOO, and EW, and we will present it in the final version of the paper. We here reply to the questions:
> > >
> > > **Question 1.**
> > >
> > > Both CAMOO and PAMOO require solving a convex optimization problem when updating the weight vector. Nevertheless, as we detail in the "Practical Implementation" subsections, we can design a scalable approach to implement it: instead of fully solving the convex optimization problem, we can take a few gradient descent steps with which we update the weight vector. The implementation of both CAMOO and PAMOO, shared in the notebook, follows this approach.
> > >
> > > We will first provide details on the compute time of CAMOO, which is less scalable compared to PAMOO. The main bottleneck in the compute time of CAMOO, assuming the Hessian is diagonal, is the need to estimate 2nd order information and the need to solve a min-max problem with dimension m, #number of objectives, and q, #number of parameters. Even though 2nd order information can be obtained with the Hutchinson method, it degrades the compute time and memory requirement. Its main bottleneck is the need to apply additional backward() operation (namely, an additional backward pass). Its compute complexity is O(N_hutchinson * backward_pass) where N_hutchinson controls the quality of approximation and is a tunable parameter. The gradient-based approach to solve min-max optimization problems require O(mq) operations. Hence, the complexity of this approach scales as O(N_hutchinson * backward_pass + mq).
> > >
> > > Unlike CAMOO, PAMOO requires solving a convex optimization problem in dimension m. The gradient has a closed form solution which has a cost of O(m²q), due to the need to calculate a matrix multiplication $J^T \cdot J$ where $J\in \mathbb{R}^{q\times m}$. Hence, to update the weight vector based on PAMOO costs O(qm²) compute steps with no backward passes. This cost can be quite negligible compared to the backward() operation which is taking place due to the need to update the network's parameters.
> > >
> > > We added experiments in which we reduce the number of update steps of the weight vector at each iteration. Our results show that PAMOO has almost the same running time as EW whereas its performance is much better. On the other hand, the running time of CAMOO is 5-6 times worse compared to EW (see AMOO Run-time section):
> > >
> > > https://docs.google.com/document/d/e/2PACX-1vRbT_IuVcnR-RHPy95d9sIuiHa07DUmjA57HvWKjjWFUjST5udLeTUygnpnQ3AvSmiuc_y-r4JpiGYI/pub
> > >
> > > Here are the running times we measured. Let N be the number of weight updates at each iteration. For the switching example in the current version of the paper, we set N=100 and N=1000 for CAMOO and PAMOO, respectively:
> > >
> > > EW-SGD: 0.408452 seconds
> > >
> > > EW-ADAM: 0.480058 seconds
> > >
> > > PAMOO-SGD: 8.654938 seconds
> > >
> > > CAMOO-SGD: 6.695782 seconds
> > >
> > > PAMOO-ADAM: 8.435489 seconds
> > >
> > > CAMOO-ADAM: 7.466246 seconds
> > >
> > > We provide additional experiments with N=10 for the switching example. For this value we get the following run times:
> > >
> > > PAMOO-SGD: 0.715199 seconds
> > >
> > > CAMOO-SGD: 3.674786 seconds
> > >
> > > CAMOO-ADAM: 3.598157 seconds
> > >
> > > PAMOO-ADAM: 0.726002 seconds
> > >
> > > We expect the gap between PAMOO and EW to reduce for larger networks; for such networks, the backward operation is significantly more demanding compared to only forward passes.
> > >
> > > **Question 2.**
> > >
> > > A lot of work has been done to establish convergence of gradient-descent in the non-convex setting, e.g., for linear neural networks (https://arxiv.org/pdf/1810.02281), NTK regime (https://arxiv.org/abs/1806.07572), and studying the inductive bias of gradient descent applied to neural networks (https://www.jmlr.org/papers/volume19/18-188/18-188.pdf), to name only a few. We believe that some of these settings can also be explored in the AMOO framework, namely, when the algorithm designer has access to approximately aligned multi-objective feedback. Nevertheless, we have not made any significant steps toward this direction and believe it is an interesting and natural next step. We will expand on this point in the conclusion section of our work. Thank you!

---

### Official Review · Reviewer_V24E · 2025-03-18

**Overall Recommendation:** 3

**Summary:**

This paper study a setting called aligned multi objective optimization (AMOO), which they define as a setting where objectives share a common solution. It studies how aligned multi-objective feedback can improve gradient convergence. The authors propose a framework for AMOO (aligned multi objective optimisation) and provide two ways of aligning multiple objectives, one by using the Hessian (CAMOO) and the other by choosing the Polyak step-size method (PAMOO). The proposed algorithms adaptively weight objectives and offer provably improved convergence guarantees. The proposed methods have been applied on SGD and ADAM and have been compared with the cost function carrying equal weightage. Experimental results show that the ADAM version of CAMOO & PAMOO shows improved convergence than the rest. Lastly, the authors point out that PAMOO is more practical than CAMOO as there is no hessian calculation involved and an approximate solution can be found by taking a few steps of projected gradient descent. The paper also proves robustness of the proposed algorithms for aligned approximately aligned objectives. The authors have experimented with a simulated setting with three loss functions. They have tested with SGD and ADAM as the optimizers of the weighted loss. These results are compared against equal-weights(EW) on the objectives. The variants suggested by the authors achieve betters score as compared to EW baseline.

**Claims And Evidence:**

The claims are supported by experiments on a simulated setting which suffice to show the advantages of the proposed algorithm to the equal weightage (EW) based baseline. The results for the simulated dataset highlight the potential of using GD algorithms designed for AMOO. However, I believe that more empirical evidence for the proposed algorithms could be more convincing. Comparison with more baseline e.g. MGDA based algorithms could be helpful. Involvement of more datasets could also strengthen the results. Empirical evidence on approximately aligned objectives could also strengthen the paper.

**Essential References Not Discussed:**

References seems to be sufficient, to the best of my knowledge

**Experimental Designs Or Analyses:**

The experimental design consists of a simulated data with a setting on three objective functions. The experiments validate the theoretical justification of the proposed using aligned MOO optimizer.

**Methods And Evaluation Criteria:**

The proposed method and the evaluation criteria does make sense for the mentioned problem.

**Other Comments Or Suggestions:**

1. Missing line numbers from the paper.
2. In algorithm 1, the updation might look like x_{k+1} = x_k - \eta g_k instead of x_{k+1} = x_k - \eta g_t

**Other Strengths And Weaknesses:**

Strengths:
1. The authors have addressed an interesting and novel problem of aligned multiple objective optimization.
2. A very well-structured and well-written paper.
3. The paper clearly describes the literature, and the differences between the proposed and existing methods, and the experimental details are easy to follow.
4. The paper is theoretically strong.
5. Convergence and robustness to alignment of algorithms have been theoretically established.

Weakness:
1. Despite writing a rich literature review, the paper misses comparison with other related works, which weakens the experimental section of the paper.
2. Empirical evidence on approximately aligned multiple objectives will be much appreciated.
3. Extensive experiments are needed to check the feasibility of these solutions in DNNs with real-world datasets.
4. Weak results and analysis section of the paper. A more elaborate explanation of the results of the results in Fig 2 will help readers to understand the results in a better way.

**Questions For Authors:**

Please see weakness section.

**Relation To Broader Scientific Literature:**

The literature on MOO considers multiple objectives in a conflicting sense. There are multiple problems in ML literature which consisting of aligned multiple objectives. Including methods like CAMOO and PAMOO will certainly be useful in a broader perspective.

**Theoretical Claims:**

Checked proof of theorem 4.2, 5.2 & E.3 in Appendix and seems well written and correct.

---

> ### Author Rebuttal · Authors · 2025-03-31
>
> Thank you for your positive feedback and recognizing the positive aspects of our work! We will address the questions that were raised in their chronological order.
>
> 1. Since prior work did not directly investigate the Aligned Multi-Objective Optimization (AMOO) setting, we focused on comparing our performance to the equal-weights baseline, which is an intuitive algorithmic approach in the multi-objective setting. We also note that recent work showed the equal-weights baseline is competitive compared to other multi-task learning algorithms on multi-task benchmarks (see https://dl.acm.org/doi/pdf/10.5555/3648699.3648908).  We also provided an extensive discussion about prior work on multi-task learning research and prior results are different from ours in Section 2.
>
> 2. Thank you for this suggestion! We include here additional results for the approximate AMOO setting in which objectives are only approximately aligned (see this anonymous link https://docs.google.com/document/d/e/2PACX-1vRbT_IuVcnR-RHPy95d9sIuiHa07DUmjA57HvWKjjWFUjST5udLeTUygnpnQ3AvSmiuc_y-r4JpiGYI/pub). We will include new results these in the final version of the paper.
>
> 3. In this work we focused on foundational work, namely, properly defining the setting, developing intuitive understanding, and theoretically grounded algorithms. Additionally, the empirical investigation we made is already outside the scope of the theory work: we test the new algorithms on non-convex optimization problems, with SGD and ADAM optimizer.  We believe our empirical investigation and open source implementation of the algorithms lay the ground for more extensive empirical investigation in different domains (multi task, LLM, RL, computer vision, etc.).
>
> 4. We will explain the results more explicitly for readers' understanding. These results reflect the Mean Squared Error (MSE) metric (that measures distance between current iterate and optimal point) of different algorithms. Interestingly,  PAMOO-ADAM, where the weights are chosen via PAMOO and the optimizer is the ADAM optimizer, has the best performance by a significant margin compared to other alternatives.
>
> Additional comments: Thank you for these comments, we will adjust the paper accordingly.

---

### Decision · Program_Chairs · 2025-05-01

**Decision:**

Accept (poster)

**Comment:**

This paper introduces Aligned Multi-Objective Optimization (AMOO), addressing scenarios where multiple objectives share a common solution. Traditional MOO focuses on conflicting objectives, but here, aligned objectives enable simultaneous optimisation. The authors propose CAMOO (curvature-aware weighting) and PAMOO (Polyak step-size-inspired weighting), both adaptively combining gradients. Theoretical guarantees show improved convergence rates over naive methods.

The reviewers have mixed feelings about the paper: while they agree that the Aligned Multi-Objective Optimization is  reasonably motivated in practice. There were concerns:  1) the analysis is restricted to the convex which might be not well connected to modern machine learing tasks; 2)  the experiments are rather toy.

Nevertheless, all the reviewers agree that the paper considered  an interesting problem and the algorithms were theoretically and empirical proved to be useful in certain settings.